# Identification and Functional Analysis of a Key Gene in the CHH Gene Family for Glucose Metabolism in the Pacific White Shrimp *Litopenaeus vannamei*

**DOI:** 10.3390/ijms26104612

**Published:** 2025-05-12

**Authors:** Chengyi Zhang, Xiaojun Zhang, Shuqing Si, Mingzhe Sun, Aixin Li, Jianbo Yuan, Fuhua Li

**Affiliations:** 1CAS and Shandong Province Key Laboratory of Experimental Marine Biology, Institute of Oceanology, Chinese Academy of Sciences, Qingdao 266071, China; a15621327187@163.com (C.Z.); 17862667253@163.com (S.S.); mzhsun@qdio.ac.cn (M.S.); 15628872790@163.com (A.L.); yuanjb@qdio.ac.cn (J.Y.); fhli@qdio.ac.cn (F.L.); 2College of Earth Science, University of Chinese Academy of Sciences, Beijing 100049, China; 3Key Laboratory of Breeding Biotechnology and Sustainable Aquaculture (CAS), Institute of Oceanology, Chinese Academy of Sciences, Qingdao 266071, China; 4Laboratory for Marine Biology and Biotechnology, Qingdao Marine Science and Technology Center, Qingdao 266071, China

**Keywords:** crustacean hyperglycemic hormone, glucose metabolism, alternate splicing variants, growth, molting

## Abstract

The crustacean hyperglycemic hormone (CHH) is a unique multifunctional neuroendocrine hormone superfamily in crustaceans, crucial for maintaining physiological homeostasis and stress adaptation. To explore the role of CHHs in shrimp metabolism and growth, we identified *LvCHH Ia*, a CHH family member who regulates glucose metabolism in the Pacific white shrimp (*Litopenaeus vannamei*), through CHH family gene classification, phylogenetic analysis, gene structure analysis, and transcription factor binding site (TFBS) prediction. Subsequently, we cloned two alternative splicing variants of this gene, *LvCHH Ia-1* and *LvCHH Ia-2*, both expressed in the nervous system but with different expression levels, and *LvCHH Ia-2* exhibiting a broader tissue distribution. Using interference (RNAi)-mediated gene silencing and recombinant protein injection, we investigated the functional similarities and differences between the two variants. Our results show that both variants affect glucose metabolism by modulating the expression of key enzyme genes involved in gluconeogenesis/glycolysis, such as *HK*, *TPI*, *PCK1*, *ALD*. Specifically, they likely regulate hemolymph glucose levels via the Wnt and PI3K-AKT signaling pathways, with *LvCHH Ia-1* exerting a more sustained effect on glucose metabolism compared to *LvCHH Ia-2*. Furthermore, LvCHH Ia may also act as a molting inhibitory hormone by suppressing the expression of ecdysone synthesis-related genes, where LvCHH Ia-2 plays a more significant role. These findings deepen our understanding of CHH regulatory mechanisms in crustaceans and provide potential applications for shrimp physiological research and aquaculture.

## 1. Introduction

The X organ/sinus gland complex (XO-SG), found in the eyestalk of decapod crustaceans, is structurally and functionally analogous to the hypothalamus–neurohypophysis system in vertebrates [1]. This complex consists of a variety of highly sensitive and specific neuroendocrine cells that can precisely detect dynamic changes in the internal environment and various signals triggered by external stimuli. In response, these cells rapidly initiate a series of complex yet orderly physiological mechanisms to maintain homeostasis within the organism [1]. The crustacean hyperglycemic hormone (CHH) family neuropeptides represent the most abundant endocrine hormone in the XO-SG of decapod crustaceans [2]. Since the initial identification of CHH, the number of CHH family members has continuously expanded. Based on their structural and functional characteristics, CHH family neuropeptides are categorized into several subtypes, including CHH, molting inhibitory hormone (MIH), gonadal inhibitory hormone (GIH), and mandibular organ inhibitory hormone (MOIH) [3,4,5]. Furthermore, according to the structural composition of genes, hormone precursors, and the characteristics of mature peptides, CHH family peptides are divided into two groups: type I and type II. Recently, a study identified the first non-crustacean member of the CHH superfamily, the ion transport peptide (ITP), as type III CHH [6]. Additionally, numerous peptides with structural similarities to CHH and functioning as biotoxins have been discovered in insects, spiders, centipedes, and even mollusks, thereby expanding the phylogenetic spectrum of CHH neuropeptides [7,8].

CHH exhibits multiple roles in the endocrine regulation of crustaceans and typically functions synergistically with other CHH family members [3,9]. For example, CHH and MIH may co-regulate molting in crustaceans [10,11]. In the Pacific white shrimp, *Litopenaeus vannamei*, CHH can inhibit ecdysone synthesis in the Y-organ by downregulating the transcription levels of enzymes involved in the ecdysone synthesis pathway as well as the ecdysone receptor EcR [12], thus affecting the regulation of the molting process [3,9,13]. Similarly, GIH and MOIH also need to collaborate with other CHH family members to regulate crustacean reproduction. For instance, in the Jonah crab (*Cancer borealis*), researchers found that CHH, MOIH, and MIH were exclusively present in the sinus glands of adult males and females, suggesting their roles in reproductive processes and development [14]. In the giant tiger prawn, *Penaeus monodon*, silencing the CHH gene altered the expression of the vitellogenin gene, indicating that changes in CHH expression can influence the synthesis and secretion of reproductive-related hormone [15].

CHH was initially named for its hyperglycemic activity [16], and this function has been confirmed across various crustacean species [15,17,18]. For instance, CHH injection significantly raises hemolymph sugar levels in crayfish (*Orconectes limosus*) and crab (*Carcinus maenas*) [19]. Conversely, in mud crab (*Scylla paramamosain*), knocking down the Sp-CHH3 gene, which encodes the type I CHH peptide, results in a marked decrease in hemolymph glucose levels [17]. Furthermore, CHH family genes generate tissue-specific and functionally distinct mRNA isoforms through alternative splicing, which has proven to be a crucial evolutionary mechanism driving functional diversification. In *P. monodon*, the two CHH gene splice variants exhibit divergent functions: dsRNA-mediated knockdown of *PmHHLPP* significantly increases hemolymph glucose levels, while *PmCHH* is involved in ovarian development and larval metamorphosis but does not directly regulate hemolymph glucose levels [20]. In swimming crab (*Portunus trituberculatus*), the splice variants *Pt-CHH1* and *Pt-CHH2* show distinct expression patterns, *Pt-CHH1* is mainly expressed in the eyestalk, brain, muscle, and hemolymph, while *Pt-CHH2* is predominantly found in the thoracic ganglia and gills, with its expression in gill tissue strongly linked to osmoregulation [21]. Alternative splicing allows crustaceans to produce transcripts with tissue-specific and functional diversity. This establishes a versatile regulatory network that meets complex physiological demands while maintaining the conserved structural integrity of the CHH gene family.

The hyperglycemic activity of CHH splice variants in crustaceans may be closely associated with the peptides’ N-terminal structure or C-terminal amidation [11,22,23]. Researchers have observed that the C-terminal regions of various CHH isoforms differ in activity and structure, with C-terminally amidated CHH exhibiting significantly higher hyperglycemic activity than its unamidated counterpart [24]. Notably, in *Panulirus japonicus*, two C-terminally amidated CHH splice variants demonstrated distinct hyperglycemic effects. The study suggested that the functional differences between *PajCHH-1* and *PajCHH-2* could be attributed to variations in five specific amino acid residues [25]. Members of the CHH family exhibit remarkable sequence conservation [26], indicating that one of the primary mechanisms influencing their functional diversity is likely the variation in amino acid composition and arrangement. Alterations in specific amino acid residues may affect CHH gene function by modifying the protein’s three-dimensional structure or enabling unique molecular recognition and interaction mechanisms. Research on *Sco-CHH* of the mud crab (*Scylla olivacea*) revealed that two mutants, rSco-CHH with missing Arg13 and Ile69 sites, completely lost their hyperglycemic activity. Additionally, the hyperglycemic response time profiles of four mutants with missing IIe2, Phe3, Asp12, and Asp60 sites differed significantly from those of the wild-type Sco-CHH [27]. This highlights the critical role of specific-position amino acids in the hemolymph glucose regulatory function of CHHs. Mutations of amino acids at certain positions can lead to partial or complete loss of CHH hyperglycemic activity. These findings provide a novel and promising research avenue for accurately predicting CHH gene function based on amino acid site analysis. It is essential to mention that CHH may also possess distinct glucose regulation capabilities through amino acid residue isomerization. It has been discovered that the Phe3 configuration of crustacean CHH can be used to differentiate it into L- or D-forms. Both isoforms exhibited clear hyperglycemic activity in the crayfish (*Pontastacus leptodactylus*), while D-Phe3-CHH induced stable and delayed hyperglycemic response kinetics. Moreover, D-Phe3-CHH exhibits either a more potent osmoregulatory activity and a more potent inhibitory activity on the molting organ during ecdysone production [28].

As pivotal neuroendocrine regulatory factor, CHH plays a core regulatory role in the physiological metabolism of crustaceans. It maintains energy homeostasis by regulating glucose metabolism, thereby influencing metabolism processes, growth, and immune responses in crustaceans [29,30,31]. Despite its importance, the exact mechanisms of CHH in glucose metabolism are not fully understood. Exploring the roles of CHH genes in this process can enhance our knowledge of shrimp’s nutritional needs and metabolic pathways, potentially improving their growth, survival rates, meat quality, and disease resistance.

In this study, we systematically classified and analyzed CHH gene family members based on previously reported *CHH* sequences from *L. vannamei* and genomic annotations from our laboratory’s dataset. Through bioinformatics analysis, we identified a CHH gene crucial for glucose metabolism regulation. Subsequently, we cloned this gene, found two distinct mRNA alternative splicing variants, and then examined their gene structure and expression patterns. To explore their functions, we used double-strand (dsRNA) interference and recombinant protein injection to knock down and overexpress the two *CHH* splicing variants, respectively. This allowed us to observe changes in hemolymph glucose levels, growth phenotypes, metabolite profiles, and the related gene expression. Our results revealed both the similarities and differences in the regulatory abilities of the two splice variants regarding processes like carbohydrate metabolism, amino acid metabolism, and molting. This study deepens our understanding of CHH’s role in glucose metabolism, providing a foundation for optimizing shrimp growth under stressful conditions, boosting aquaculture productivity, and informing feed development and management strategies.

## 2. Results

### 2.1. Identification and Classification of CHH Gene Family Members

A total of 143 CHH sequences were collected from our *L. vannamei* genome database and publicly accessible databases. Upon analyzing the domain architecture, it was found that 136 of these sequences (Appendix A) possessed the signature domain (Crust_neurohorm) characteristic of the CHH family gene (Figure 1A). To gain a deeper understanding of the genomic distribution of the CHH genes within the *L. vannamei* genome, a chromosome localization map was constructed. The results revealed that these CHH sequences were spread across 29 genes on eight scaffolds (Figure 1B). Further examination through chromosome localization demonstrated that most of these genes were arranged in clusters. Specifically, the CHH gene family is composed of five gene clusters: the LVANscaffold_1036 gene cluster contains five CHH genes, LVANscaffold_1388 encompasses two tandem duplication genes, both LVANscaffold_2640 and LVANscaffold_2938 each contain a gene cluster of three CHH genes, and LVANscaffold_1388 exhibits the most significant expansion, forming a large gene cluster with a total of 13 genes. Sequences within these clusters display significant similarity (Appendix A), suggesting that the proliferation of CHH genes might have arisen from local tandem gene duplication events. Subsequently, a phylogenetic analysis of these genes’ protein sequences was conducted (Appendix A). Based on the resulting phylogenetic tree and previous identification using CDS sequences, all the CHH family genes were classified into three types: Type I, Type II, and Type III (Figure 1C). Type I was further subdivided into three subtypes: CHH Ia, CHH Ib, and CHH Ic, Type II comprises MIH/MOIH/GIH/VIH, while type III corresponds to ITP. Notably, Type Ib CHH clusters with type II and type III CHHs, forming a distinct branch separate from type Ia and type Ic on the phylogenetic tree. Moreover, type Ia CHH consists of a single gene with no evidence of gene expansion, in contrast to type Ic CHHs, which have undergone considerable expansion, incorporating 18 CHH genes.

### 2.2. Structura Characterization of CHH Gene Family Members

A prediction motif analysis was performed on the CHH sequences presented above (Figure 2A). The results reveal that all the CHH proteins contain motifs 1, 2, and 6, which are highly conserved within the CHH gene family. The motifs were compared to the known functional protein motifs contained within the Expasy Prosite database (Figure 2B). The results show that motif 2 is characteristic of the arthropod CHH/MIH/GIH neurohormone family [4]. Motif 1 encompasses the sequence pattern site of the eukaryotic pancreatic hormone family and may significantly influence glucose metabolism and feeding behavior [32]. Motif 6 features a distinctive functional location of the carbohydrate kinase family PfkB, which specializes in carbohydrate substrates such as fructose, ribose, adenosine, and inositol and regulates carbohydrate metabolism [33]. Motif 3 is present in both type I and type II CHHs, possessing a similar motif in the mammalian lipolytic enzyme “GDXG” family, potentially acting as a serine active site involved in lipid metabolism [34]. Motif 7, a defining feature of type II CHHs, shares homology with crustacean hemocyanin and insect larval serum protein functional motifs, suggesting its potential role in regulating oxygen transport, free radical scavenging, and molting growth in crustaceans [35,36].

To further investigate the classification and evolutionary relationships of the CHH gene family in *L. vannamei* and identify the key CHH genes involved in glucose metabolism regulation in shrimp, we analyzed the 136 CHH sequences with characteristic CHH family domains, and seven CHH sequences from *L. vannamei* [37], *P. monodon* [20], *Scylla paramamosain* [17], *Scylla olivacea* [27], *Neohelice granulate* [38], and *Panulirus japonicus* [25]. These seven proteins have been reported to be involved in the regulation of hemolymph glucose (Appendix A). We constructed a phylogenetic tree based on CHH sequences potentially associated with glucose metabolism in crustaceans (Figure 3A). The results revealed that 12 CHH sequences were clustered within the type Ia CHH subgroup, including two previously reported *L. vannamei* CHHs known to regulate glucose metabolism (Genebank: AAN86056.1 and AAN86057.1) [37]. Other known crustacean CHHs involved in glucose metabolism are also grouped within the Type Ia CHH subgroup. Based on these findings, we propose that the primary CHH family gene regulating glucose metabolism likely belongs to the Ia type CHH subgroup.

To verify whether *CHH Ia* is a key gene for glucose metabolism, we predicted the transcription factor binding sites (TFBSs) within 2000 bp upstream of the transcription start site of the *CHH Ia* gene in *L. vannamei*. Using a threshold of *p*-value < 10^–7^, we identified 39 TFBSs, which fell into six functional categories: glucose metabolism, cell proliferation and differentiation, nervous system repair and development, immune processes, gonadal development, and protein metabolism (Figure 3B). Among these, seven transcription factors associated with glucose metabolism were detected: CREB1, CREBBP, GATA4, TP53, RCOR1, KDM1A, and PPARG. These factors are implicated in pathways such as glycolysis inhibition, cAMP/cGMP signaling, thyroid hormone signaling, PI3K-Akt signaling, glucagon signaling, and AMPK signaling. The TFBS prediction results further confirm that the *CHH Ia* gene is closely associated with glucose metabolism processes.

### 2.3. Gene Structure of LvCHH Ia

Based on the integrated analysis of phylogenetic, motif, and TFBS prediction results, we propose that type Ia CHH is crucial in regulating glucose metabolism in *L. vannamei*. This gene is designated as *LvCHH Ia*. Comparative sequence analysis of the eleven proteins classified under the CHH Ia subgroup (Appendix A) revealed high sequence conservation, and they can primarily be divided into two types by amino acid differences at the C-terminus. The presence of a single gene of this type in the genome suggests that the two distinct protein sequences are generated through alternative splicing of the *LvCHH Ia* gene, potentially fulfilling distinct physiological functions. Using the mRNA sequence of the *LvCHH Ia* gene (Appendix A), we successfully amplified two distinct alternative splicing variants from the eyestalk cDNA of *L. vannamei*, which were named *LvCHH Ia-1* and *LvCHH Ia-2*, respectively. We comprehensively analyzed the nucleic acid sequence of the *LvCHH Ia* gene and its two alternative splicing variants. The result revealed that the *LvCHH Ia* gene consists of four exons. The *LvCHH Ia-1* variant is a short splicing form comprising exons I, II, and IV. In contrast, the *LvCHH Ia-2* variant is a long splicing form, encompassing exons I, II, III, and IV. Notably, a stop codon sequence is located at the end of exon III, leading to premature translation termination of *LvCHH Ia-2* (Figure 4A). Both LvCHH Ia proteins share a conserved structure, consisting of a signal peptide (1–30), a leader peptide (31–67), a KR disalt bond (67–68), and a mature peptide (70-141/142). They possess six cysteines forming three pairs of disulfide bonds, which are characteristic of the CHH family (Figure 4B). The divergence between the two protein sequences begins at amino acid 114. Using the GPS-PALL algorithm, we predicted lysine acetylation at the C-terminus of *LvCHH Ia-1*; however, no such modification was predicted in *LvCHH Ia-2* (Appendix A). Spatial structure prediction results showed that the tertiary structures of *LvCHH Ia-1* and *LvCHH Ia-2* are predominantly folded by α-helix and irregular coil through intermolecular interactions (Figure 4C,D). In predicting small molecule probe interaction sites on the three-dimensional structures of *LvCHH Ia-1* and *LvCHH Ia-2*, eight consensus sites (CSs) of ligand-binding hot spots were identified (Figure 4C). Specifically, the 000 CS of *LvCHH Ia-1* comprises 28 probe clusters, whereas the 000 CS of *LvCHH Ia-2* includes 21 probe clusters (Figure 4D). Among the remaining CSs with analogous positions, *LvCHH Ia-1* exhibits more probe clusters than CHH Ia-2. Additionally, *LvCHH Ia-2* features two distinctive CSs located on the terminal α-helix at the C-terminal end.

### 2.4. Gene Expression Patterns of LvCHH Ia

To visualize the expression patterns of CHH family genes across various tissues in shrimp, a heat map was constructed using our previous transcriptome data from 16 tissues (Figure 5A). Notably, the type Ia CHH gene *CHHS2640_4* showed significantly elevated expression levels in the eye stalk, brain, thoracic ganglion, and ventral nerve. The type Ib CHH gene *CHHS3427*(*LVAN22614*) exhibits high expression levels in the intestine, eye stalk, brain, thoracic ganglia, and ventral nerves. *CHHS2938* (*LVAN19742/LVAN19744/LVAN19748*) predominantly expressed in the thoracic ganglion. The type Ic CHH gene *CHHS2916* (*LVAN19516*) was detected to be expressed in the eye stalk and ventral nerve. Other type Ic CHH genes, such as *CHHS2490* (*LVAN16174-LVAN16186*), showed low expression in organs apart from the eye stalk and ventral nerve. The type II CHH genes *VIHC 1028-1*, *VIHC 1028-2*, *MIHC 1028*, and *MIHS1036-3* are mainly expressed in the brain, thoracic ganglion, and eye stalk, respectively.

The expression distribution of the two alternative splicing variants of the *LvCHH Ia* gene was verified in 13 tissues. The results revealed distinct expression patterns among the isoforms of *LvCHH Ia*. Specifically, *LvCHH Ia-1* showed high expression in the gill, thoracic ganglion, brain, and ventral nerve tissues. In contrast, *LvCHH Ia-2* was mainly expressed in the thoracic ganglia, intestinal tract, muscle, gills, and brain (Figure 5B). The differences in experimental shrimp batches, growing duration, and experimental conditions between this study and previous transcriptome may influence the tissue-specific expression distribution. Overall, the expression profiles of the two *LvCHH Ia* variants across various tissues were generally consistent with the transcriptome data, with both showing significant expression levels in the thoracic ganglia. It is worth noting that *LvCHH Ia-2* displays a more widespread tissue distribution than *LvCHH Ia-1*.

### 2.5. LvCHH Ia Gene RNA Interference

In the RNAi pre-experiment, compared to the dsEGFP control group, the relative expression levels of *LvCHH Ia-1* and *LvCHH Ia-2* in the thoracic ganglion were significantly decreased in both treatment groups (Appendix A). The optimal dsRNA injection dosage for both groups was 2 μg/individual, achieving interference efficiencies of 80.9% and 77.3%, respectively. The optimal injection dose was calculated as 0.35 μg/g body weight.

In the dsEGFP group, the hemolymph glucose level was higher than that in the uninjected dsCHH Ia-1, and dsCHH Ia-2 groups 48 h post dsRNA injection. Conversely, the hemolymph glucose levels in the dsCHH Ia-1 and dsCHH Ia-2 groups were lower than in the uninjected group, with the dsCHH Ia-1 group exhibiting a lower level than the dsCHH Ia-2 group. These results suggest that RNAi-mediated knockdown of dsCHH Ia-1 and dsCHH Ia-2 suppressed the increase in hemolymph glucose, with dsCHH Ia-1 showing a more pronounced inhibitory effect than dsCHH Ia-2 (Figure 6C).

The formal RNAi experiment lasted for 23 days. At the end of the experiment, compared to the PBS and dsEGFP control groups, the dsCHH Ia-1 and dsCHH Ia-2 groups showed significant suppression of CHH gene expression in the thoracic ganglion, with interference efficiencies reaching 98.5% and 85.7%, respectively (Figure 6A,B). To assess whether the glucose metabolism regulation pathway could effectively modulate hemolymph glucose levels by activating the second messengers cAMP and cGMP, thereby achieving cascade amplification of regulatory signals, we measured their concentrations in the hemolymph after the RNAi experiment. Results showed that in both treatment groups, cAMP and cGMP levels in the shrimp were significantly lower than in the dsEGFP group. Following *LvCHH Ia-2* knockdown, the cAMP and cGMP concentrations in shrimp hemolymph changed markedly compared to the control group but the levels significantly lower than in the dsCHH Ia-1 group (Figure 6D,E).

To investigate the effects of *LvCHH Ia-1* and *LvCHH Ia-2* interference on the growth of *L. vannamei*, we collected data on body weight and body length before and after the RNAi experiment. The results indicated that following 23 days of sustained RNAi, the body weight and body length in the dsCHH Ia-1 group were significantly reduced compared to those in the PBS and dsEGFP groups, suggesting that the knockdown of *LvCHH Ia-1* inhibited shrimp growth. In contrast, the dsCHH Ia-2 group exhibited greater weight gain compared to the PBS and dsEGFP groups (Figure 6F), whereas their increase in body length was slightly lower than that observed in the other two groups (Figure 6G). Although a trend toward enhanced weight gain was observed in the dsLvCHH Ia-2 group by the end of the experiment, statistical analysis revealed that the difference was not statistically significant (*p*-value > 0.05).

By recording the molting numbers of shrimp in each group during the experiment, we found that injecting two types of dsCHH Ia increased molting frequency in all (Figure 6H). As a multifunctional neuropeptide hormone, LvCHH Ia may act as an ecdysone inhibitor in shrimp. After RNAi-mediated knockdown, the inhibitory effect is alleviated, stimulating ecdysone-related pathways. The elongated splice variant *LvCHH Ia-2* may exert a more pronounced ecdysone inhibitory effect.

### 2.6. Transcriptome Analysis After LvCHH Ia Knockdown

This study generated 117.71 Gb of raw reads from 18 transcriptome sequencing libraries, with Q30 values ranging from 96.69% to 97.72%, indicating high-quality sequencing data (Appendix A). All raw reads were deposited at the NCBI Sequential Read Archive (SRA) (PRJNA1240581). Gene expression levels were determined by calculating FPKM values for each sample. The squared Pearson correlation coefficients (R^2^) between thoracic ganglion and hepatopancreas samples were above 0.937 and 0.939 (Figure 7A). Principal component analysis (PCA) revealed distinct separation of samples from different tissues (Appendix A), reflecting the high sample consistency and repeatability.

In the transcriptome analysis, with the dsEGFP group as the control, the dsCHH Ia-1 group exhibited a much higher number of differentially expressed genes (DEGs) in the hepatopancreas than the dsCHH Ia-2 group. Specifically, the dsCHH Ia-1 group had 1733 DEGs, of which 920 were upregulated and 813 were downregulated. In contrast, the dsCHH Ia-2 group had 1189 DEGs, comprising 723 upregulated and 466 downregulated genes. The two groups shared 413 DEGs in the hepatopancreas. However, in the thoracic ganglion, the dsCHH Ia-1 group exhibited significantly fewer DEGs than the dsCHH Ia-2 group, with 1196 DEGs (599 upregulated and 597 downregulated). Conversely, the dsCHH Ia-2 group identified 1895 DEGs, including 751 upregulated and 1144 downregulated genes. The two groups shared 560 DEGs in the thoracic ganglion (Figure 7B).

Following the dsRNA-mediated knockdown of the *LvCHH Ia gene*, a heatmap was generated using the FPKM values of the aforementioned transcription factors and randomly selected background transcription factors based on transcriptomic data (Figure 3C). Among these, only CREB exhibited decreased expression upon gene interference, suggesting that CREB may serve as a key transcription factor involved in CHH-mediated regulation of glucose metabolism.

#### 2.6.1. Differential Expressed Genes Analysis in Thoracic Ganglia

Gene Ontology (Go) enrichment analysis of differentially expressed genes (DEGs) in the thoracic ganglia showed that, in the dsCHH Ia-1 and dsCHH Ia-2 groups, DEGs were mainly enriched in the “biological process” related to chromosome structural dynamic and DNA replication. Within the “cellular component” category, DEGs were primarily localized in the nucleus’s chromosomes, nucleosomes, and DNA–protein complexes. Regarding “molecular function”, DEGs exhibited significant enrichment in binding activities related to transmembrane transporter activity, neurotransmitter transport activity and phosphotransferase activity (Figure 7C). Kyoto encyclopedia of genes and genomes (KEGG) pathway analysis further indicated that the common DEGs were significantly enriched in several key pathways, including DNA replication, Wnt signaling pathway, MAPK signaling pathway, phagosome pathway, and glutathione metabolism pathway (*p*-value < 0.05) (Figure 7C).

GO enrichment analysis of the DEGs specific to the dsCHH Ia-1 group revealed that these genes were primarily enriched in “biological process” such as cell adhesion and carbohydrate metabolism. In terms of “cellular component”, the DEGs were mainly localized to membrane-associated structures, including cytoplasmic vesicles and the plasma membrane. Regarding “molecular function”, significant enrichment was observed in activities related to oxidoreductase, calcium ion binding, and protein binding involving transcription factor interactions (Figure 7D). Furthermore, KEGG pathway analysis indicated that these DEGs were predominantly enriched in pathways associated with fatty acid degradation, amino acid biosynthesis, and glycolysis/gluconeogenesis (*p*-value < 0.05) (Figure 7D).

GO enrichment analysis of the DEGs specific to the dsCHH Ia-2 group revealed that, in the “biological process” category, these genes were primarily enriched in small molecule biosynthetic processes, the movement of cellular and subcellular components, ion transport, regulation of cell signaling, protein phosphorylation, and the modulation of responses to external stimuli. In the “cellular component” category, the DEGs were mainly localized to the extracellular region, plasma membrane, ribosomes, and ribonucleoprotein complexes. In terms of “molecular function”, significant enrichment was observed in protein kinase activity, phosphotransferase activity, microtubule binding, and endopeptidase regulatory activity (Figure 7E). KEGG pathway analysis further demonstrated that these DEGs were significantly enriched in pathways such as the phosphatidylinositol signaling system, ribosome, endocytosis, the TGF-β signaling pathway, and inositol phosphate metabolism (*p*-value < 0.05) (Figure 7E). Among these, all 16 genes involved in the phosphatidylinositol signaling pathway were downregulated (Appendix A), whereas genes associated with the TGF-β signaling pathway were upregulated.

The differential gene expression induced by dsCHH Ia-1 and dsCHH Ia-2 in the thoracic ganglia profoundly influenced a diverse array of biological processes, including the transmission of genetic information, signal transduction, molecular transport, and antioxidant defense mechanisms.

#### 2.6.2. Differential Expressed Genes Analysis in Hepatopancreas

Enrichment analyses were conducted on the DEGs identified in the hepatopancreas of the dsCHH Ia-1 and dsCHH Ia-2 groups. GO enrichment revealed that the common DEGs between the two groups were predominantly involved in various metabolic processes within the “biological process” category, including carbohydrate metabolism, hexose metabolism, carboxylic acid metabolism, small molecule metabolism, cellular amino acid metabolism, organic acid metabolism, oxidative acid metabolism, and redox processes. Within the “cellular component” category, these genes were primarily localized in the cytoplasm. In terms of “molecular function”, there was significant enrichment in enzymatic activities such as oxidoreductase, protein kinase, phosphotransferase, and peptidase, as well as in carbohydrate-binding activity (Figure 7F). Further KEGG pathway enrichment analysis revealed that these DEGs were significantly associated with 14 metabolism-related pathways (*p*-value < 0.05), including glycolysis and gluconeogenesis, tyrodine metabolism, glycerophospholipid metabolism, cytochrome P450-mediated metabolism, the FOXO signaling pathway, and glutathione metabolism (Figure 7F). Notably, 14 genes were commonly enriched in the glycolysis/gluconeogenesis and carbon metabolism pathways (*p*-value < 0.05). Several key enzymes involved in gluconeogenesis, such as phosphoenolpyruvate carboxykinase 1(*PCK1*) and acetyl-CoA synthetase (*ACS*) were downregulated, whereas genes encoding enzymes involved in glycolysis, including hexokinase (*HK*), triosephosphate isomerase (*TPI*), aldehyde dehydrogenase (*ALDH*), and fructose-bisphosphate aldolase (*FBA*), were upregulated (Appendix A).Taken together, the GO and KEGG enrichment analyses suggest that *LvCHH Ia-1* and *LvCHH Ia-2* may coordinately participate in modulating metabolic processes in the shrimp hepatopancreas. On one hand, by modulating key metabolic pathways such as glycolysis and gluconeogenesis, they may fine-tune intracellular glucose metabolic flux and energy supply to maintain cellular energy homeostasis. On the other hand, they appear to be involved in various metabolic processes, including amino acid and glycerophospholipid metabolism, thereby regulating the synthesis and degradation of diverse biomolecules essential for normal cellular functions.

GO enrichment analysis of the DEGs uniquely expressed in the dsCHH Ia-1 group revealed significant enrichment in several biological processes, particularly aminoacyl-tRNA activation, carbohydrate metabolism, amino acid metabolism, and various acid metabolic processes, including those involving carboxylic acids, oxoacids, and organic acids. In the “cellular component” category, these genes were predominantly localized to membrane-associated structures, such as vesicle membranes and the Golgi membrane. Regarding “molecular function”, there was notable enrichment in enzymatic activities, including oxidoreductase activity, aminoacyl-tRNA ligase activity, hydrolase activity, as well as iron ion binding (Figure 7G). Further KEGG pathway analysis demonstrated that these unique DEGs were significantly enriched in pathways such as aminoacyl-tRNA biosynthesis, other glycan degradation, and starch and sucrose metabolism (*p*-value < 0.05). In addition, 13 other pathways related to carbohydrate metabolism and protein biosynthesis were also significantly enriched (*p*-value < 0.05), including carbon metabolism, multiple amino acid metabolic pathways, fatty acid metabolism, and cytochrome P450-mediated metabolism (Figure 7G). Within the starch and sucrose metabolism pathway, seven alpha-amylase A genes and one hexokinase gene were found to be upregulated. Collectively, these findings suggest that the dsCHH Ia-1-specific DEGs are primarily enriched in pathways associated with material and energy metabolism (e.g., carbohydrate and amino acid metabolism), protein biosynthesis (including aminoacyl-tRNA activation and related enzymatic activities), and redox homeostasis (via oxidoreductase activity). The subcellular localization of these genes to vesicle and Golgi membranes further implies potential involvement in intracellular transport and secretion. Therefore, these genes are likely to play essential roles in regulating shrimp growth and development, nutrient utilization, immune defense, and the maintenance of internal physiological homeostasis.

To elucidate the function of *LvCHH Ia-2*, enrichment analyses were performed on the DEGs uniquely identified in the dsCHH Ia-2 group. GO enrichment analysis revealed that these specific DEGs were primarily associated with various metabolic processes within the “biological process” category, including drug metabolism, chitin metabolism, amino sugar metabolism, amino glucoside compound metabolism, and carbohydrate metabolism, as well as processes related to cell adhesion. In the “cellular component” category, the DEGs were predominantly localized to the extracellular region and cytoskeleton. Regarding “molecular function”, these genes were significantly enriched in molecular binding activities such as chitin binding and pyridoxal phosphate binding, as well as catalytic activities including serine-type peptidase and NADH dehydrogenase activity (Figure 7H). We identified twenty-eight DEGs associated with chitin metabolism that were upregulated (Appendix A). These genes included chitinase (*CHI*) and chitotriosidase (*CHITL*), which are involved in chitin degradation and clearance, as well as extensin-like protein (*FN*), chondroitin proteoglycan (*CPG*), and cartilage oligomeric matrix protein (*COMP*), which are essential for bone development. KEGG pathway analysis further revealed that these group-specific DEGs were significantly enriched in nine pathways related to cellular metabolism, immune defense, and homeostasis. These included phagosome, amino sugar and nucleotide sugar metabolism, sphingolipid metabolism, lysosome, glycerophospholipid metabolism, motor protein function, peroxisome, cysteine, and methionine metabolism, and oxidative phosphorylation (*p*-value < 0.05) (Figure 7H).Overall, the integrated results from GO and KEGG enrichment analyses suggest that the DEGs unique to the dsCHH Ia-2 group are mainly involved in processes related to metabolism, cellular structure and adhesion, molecular binding and catalysis, chitin degradation, skeletal development, immune defense, and physiological homeostasis. Collectively, *LvCHH Ia-2* is likely to contribute to shrimp growth and development, nutrient and energy metabolism, immune protection, and the maintenance of internal homeostasis.

### 2.7. rLvCHH Ia Protein Injection

Recombinant mature peptides of *LvCHH Ia-1* and *LvCHH Ia-2* were successfully purified via the E. coli prokaryotic expression system (Figure 8A), and overexpression experiments were subsequently performed for protein injection. Hemolymph glucose concentration measurements were conducted on one-hour post-injection of rLvCHH Ia proteins. The results showed that injection of 100 pmol of recombinant protein exhibited the most pronounced hyperglycemic effect. This demonstrated that this dosage represents the optimal concentration for protein injection (Appendix A).

In the formal recombinant CHH injection overexpression experiment, the glucose concentration in the hemolymph of shrimp in the PBS, rLvCHH Ia-1, and rLvCHH Ia-2 injection groups increased significantly by one hour post injection. Notably, the hemolymph glucose increase caused by the PBS injection was approximately half that of the two CHH protein injections. Additionally, the hemolymph glucose levels in the shrimp gradually decreased over time following injection and returned to the baseline level after 6 h. It is worth noting that the rLvCHH Ia-1 exhibited a more prolonged hyperglycemic effect. These findings indicate that external stimulation (via injection) leads to an elevation in hemolymph glucose levels in shrimp. The pronounced difference between the rLvCHH Ia groups and the control group confirms that both proteins possess significant hyperglycemic activity, albeit with varying efficacy (Figure 8B).

## 3. Discussion

### 3.1. Evolution Analysis of the CHH Gene Family

The crustacean CHH family shows significant evolutionary conservation but functional diversity across species. Paralogous gene emergence via tandem repeat or whole genome duplication may drive this diversification, which is a common evolutionary mechanism [13]. Tandem repeat is a major gene family expansion driver, and genes in clusters often gain new functions through replication and mutation, enabling environmental stressors [39,40,41]. In *L. vannamei*, the CHH gene cluster expansion may relate to the precise regulation of metabolism, molting, reproduction, and growth. The significant expansion of type Ic CHH genes is likely reflecting adaptation to environmental challenges like salinity and temperature fluctuations. As a low-expression gene, CHH Ic genes may face minimal selection pressure, increasing duplication events’ retention likelihood [42]. Phylogenetic analysis shows the highly expressed CHH Ia gene as a separate branch without significant expansion (Figure 1C). Natural selection may maintain a single CHH Ia copy to prevent dose overload induced adaptive costs, as its expression changes could disrupt key metabolic pathways due to its fundamental role in basal metabolism [43,44]. Phylogenetic analysis indicates that type Ib CHHs cluster with type II and type III CHHs, while type Ia and type Ic CHHs form distinct clades. This suggests that type I CHH genes may represent an early divergent lineage within the gene family. Type Ia CHH retains its ancestral structure and function without expansion, while type Ic and type Ib CHHs have expanded widely, gaining adaptive mutations. Type II and type III CHH genes are functionally specialized and diverged during evolution. Specifically, type II CHHs regulate molting and reproduction, and type III CHH gene maintains ion homeostasis. Based on evolutionary analysis, CHH Ia is proposed as the most likely ancestral family gene. Shrimp experienced specific environmental pressures throughout the evolutionary process, leading to CHH Ib and CHH Ic genes differentiation, which adapted to changing environments through extensive gene expansions. Consequently, functionally specialized type II and type III CHH genes emerged due to shrimp survival strategies in various niches.

### 3.2. Effect of LvCHH Ia on Hemolymph Glucose

In the *LvCHH Ia* RNAi and recombinant protein injection experiments, injection stimulation significantly altered shrimp hemolymph glucose levels, likely due to the stress-activated metabolic pathway elevating glucose. In the RNAi experiment, the dsCHH Ia-1 and dsCHH Ia-2 groups had much lower hemolymph glucose levels than the dsEGFP group, indicating that *LvCHH Ia-1* and *LvCHH Ia-2* knockdown inhibits short-term stress response glucose metabolism activation (Figure 6C). Furthermore, one hour post recombinant protein injection, the rLvCHH Ia-1 and rLvCHH Ia-2 groups’ hemolymph glucose levels rose markedly (Figure 8B), showing that LvCHH Ia recombinant protein injection effectively elevated hemolymph glucose levels, possibly in a dose-dependent manner.

The results demonstrate that rLvCHH Ia-1 and rLvCHH Ia-2 could regulate shrimp hemolymph glucose. Previous studies found phenylalanine (F) at position 3 and arginine (R) at position 13 in the mud crab (*Scylla olivacea*) CHH mature peptide are key to glucose metabolism [27]. Our study found that these two amino acid sites are conserved in *LvCHH Ia-1* and *LvCHH Ia-2* (Figure 4B). Notably, the short splice variant *LvCHH Ia-1* exhibits a more sustained glucose increasing effect than the long splice variant *LvCHH Ia-2*. This hemolymph glucose regulation difference may arise from their C-terminus amino acid residue sequence variations [24,45]. Furthermore, C-terminal amidation modification has been demonstrated to be significant for CHH family peptides. This post-translational modification improves protein stability and signaling pathway activation, thus enhancing hemolymph glucose management [46]. In the narrow-clawed crayfish, *Astacus leptodactylus*, amidated Asl-rCHH regulated hemolymph glucose more effectively than unmediated Asl-rCHH-Gly in bioassays [47]. In our study, we predicted a lysine amidation modification signal at *LvCHH Ia-1*’s C-terminus, but not in *LvCHH Ia-2* (Appendix A). So, C-terminal amidation modification likely contributes to the hemolymph glucose regulation differences between the two *LvCHH Ia* splice variants.

### 3.3. Mechanisms of LvCHH Ia Affecting Glucose Metabolism

After *LvCHH Ia* knockdown, significant changes in the expression levels of glycolysis/gluconeogenesis pathway were observed in the hepatopancreas of both groups (Figure 7F). The DEGs enriched in gluconeogenesis, such as *PCK1* and *ACS* (key enzymes in gluconeogenesis), were markedly suppressed. *PCK1* is a rate-limiting enzyme for gluconeogenesis, which synthesizes glucose or glycogen from non-carbohydrate sources to maintain hemolymph glucose balance [48]. *ACS* synthesizes from acetate and CoA, indirectly promoting gluconeogenesis by activating pyruvate carboxylase (*PC*) [49]. Simultaneously, after *LvCHH Ia-1* and *LvCHH Ia-2* knockdown, key enzyme genes in the glycolysis and carbon metabolism pathway, such as *HK*, *TPI*, *FBA* and *ADLH*, were upregulated. *HK* acts as the initial rate-limiting enzyme in glycolysis and catalyzes glucose phosphorylation to provide high-energy phosphate intermediates for subsequent metabolic reactions [50]. *TPI* interconverts dihydroxyacetone phosphate (*DHAP*) and glyceraldehyde 3-3-phosphate (*G3P*), which are critical for glycolysis and gluconeogenesis [51,52]. ***PFK*** converts fructose-6-phosphate into fructose-1,6-bisphosphate, a key regulatory step in glycolysis [53]. Furthermore, in the dsCHH Ia-1 group, α-amylase-like (*α-AMY*) genes, which hydrolyze starch and glycogen, were upregulated, along with 10 glycosidic bond hydrolases related to polysaccharide metabolism (Appendix A). Silencing the two *LvCHH Ia* splice variants suppresses gluconeogenesis, prompting cells to use glycolysis for rapid energy production upon environmental stimuli. This means *LvCHH Ia* can directly modulate shrimp hemolymph glucose by regulating gluconeogenesis and glycolysis. Following the disruption of *LvCHH Ia*, the upregulated hydrolysis glycolysis genes may compensate for energy deficits from low hemolymph glucose. Hemolymph glucose measurements confirm *Lv*CHH Ia’s key role in glucose metabolism and homeostasis. Notably, dsCHH Ia-1 has more significantly enriched genes in glycolysis/gluconeogenesis and various glucose-producing compensatory mechanisms than dsCHH Ia-2, which may explain LvCHH Ia-1’s superior regulatory capacity.

In our study, the disruption of *LvCHH Ia* significantly impacted signaling pathways in the thoracic ganglion. Enrichment analyses from both the dsCHH Ia-1 and dsCHH Ia-2 groups indicated that *LvCHH Ia*, functioning as an endocrine hormone gene, is critical for maintaining internal homeostasis in shrimp, and its knockdown may result in systemic metabolic dysregulation. Functional enrichment of the DEGs shared between the two treatment groups revealed their involvement in core cellular processes, such as DNA replication and chromosomal structural modifications. This suggests a conserved role for both splice variants in fundamental cell biology, particularly in maintaining genomic stability, facilitating genetic information transmission, and regulating cell cycle progression. Moreover, enrichment of transmembrane and neurotransmitter transport activity implies a shared role of these variants in modulating neuronal function, consistent with their classification as neuropeptides. Notably, the common DEGs in the thoracic ganglion were enriched in the Wnt signaling pathway (Figure 7C). In mammals, this pathway is activated by cortical neurons to enhance glycolysis and meet increased energy demands, thereby promoting glucose utilization [54,55]. In our findings, CHH interference resulted in the upregulation of seven genes involved in the Wnt pathway, suggesting that CHH knockdown may activate the Wnt signaling pathway in the thoracic ganglion, thereby stimulating glycolysis and restoring glucose homeostasis.

The DEGs uniquely expressed in the dsCHH Ia-1 group highlighted biological processes related to intercellular communication (cell adhesion) and energy metabolism (carbohydrate metabolism), indicating a distinctive role in tissue morphogenesis, cellular signaling, and energy supply. In contrast, DEGs specific to the dsCHH Ia-2 group were associated with a variety of intracellular and extracellular processes, including small molecule biosynthesis, regulation of cell signaling, and protein phosphorylation. KEGG analysis revealed that DEGs enriched in the phosphatidylinositol signaling pathway were uniformly downregulated (Figure 7D). Interference with *LvCHH Ia-2* promoted the cascade amplification of the PI3K-Akt signaling pathway, as evidenced by the downregulation of three key genes—phosphatidylinositol 3,4,5-trisphosphate 3-phosphatase (*PTEN*), inositol polyphosphate-4-phosphatase type I A (*INPP4A*), and inositol polyphosphate 5-phosphatase (*INPP5*)—which act as negative regulators of this pathway [56,57,58]. The alleviation of their inhibitory effect on AKT led to the activation of downstream glycolysis and suppression of glycogen synthesis, thereby maintaining glucose homeostasis (Figure 7E).

In insects, neuropeptide F (NPF)—a homolog of vertebrate pancreatic hormones—has been shown to activate PI3K and CREB downstream signaling, indirectly regulating glucose metabolism and feeding behavior [32].

The difference in glucose metabolism regulatory capacity between *LvCHH Ia-1* and *LvCHH Ia-2* may be due to amino acid variations, alternative splicing and C-terminal amidation modification. When predicting structurally active binding sites for LvCHH Ia, we found that both splice variants have eight consensus sites (*CSs*). However, *LvCHH Ia-1* has more probe clusters than *LvCHH Ia-2*, implying that *LvCHH Ia-1* possesses more binding sites and, thus, better regulation of glucose metabolism. Specifically, the terminal α-helix of *LvCHH Ia-2* was predicted to contain two potential binding sites within the *LvCHH Ia* gene, indicating that the long splice isoform of *LvCHH Ia* may have additional functions beyond glucose metabolism.

### 3.4. Effects of LvCHH Ia on Molting and Growth

In RNAi experiments, both dsCHH Ia-1 and dsCHH Ia-2 groups showed higher molting frequency than the control, and molting quantity of dsCHH Ia-2 group higher than that of dsCHH Ia-1 group (Figure 6H). GO and KEGG enrichment analyses revealed that DEGs in the dsCHH Ia-2 group were predominantly enriched in the carbohydrate metabolism, phagosomes, chitin metabolic, amino sugar metabolic, and TGF-β signaling pathway (Figure 7E). Specifically, 28 chitin metabolic related DEGs were upregulated(log2FoldChange > 1), including *CHI*, balbiani ring protein 3(a chitin-like protein in insects), Chondroitin Proteoglycans *CPGs*, and *COMP*. *CHI* is crucial for chitin degradation in crustaceans by cleaving β-1,4-glycosidic bonds [59]. *CPGs* may aid old exoskeleton breakdown and new exoskeleton synthesis [60], while *COMP* may ensure new exoskeleton integrity by regulating cell processes. Moreover, peritrophin-A, a chitin-binding domain protein prevalent in insect exoskeleton [61], was markedly enriched. The phagosome pathway enriched 14 upregulated genes like *COMP*, integrin β subunit, and tubulin α/β subunits, which may help remove old exoskeleton metabolites and build the new one. These findings suggest that *LvCHH Ia* may play a regulatory role in the molting pathway of *L. vannamei*. Specifically, *LvCHH Ia-2* is likely to contribute to the coordination of exoskeletal degradation, remodeling, and regeneration during ecdysis, potentially by repressing the chitin metabolic pathway. As a key gene involved in the regulation of molting, *LvCHH Ia-2* appears to be essential for maintaining normal developmental progression and morphological renewal in shrimp.

In the continuous RNAi experiment, the dsCHH Ia-1 group had significantly lower body weight than the PBS control (Figure 6F), indicating the *LvCHH Ia*’s role in regulating shrimp growth. Our study shows that *LvCHH Ia-1* mainly modulates glucose metabolism by enhancing gluconeogenesis, providing direct energy substrates for growth and supporting energy-dependent processes such as protein synthesis and tissue repair. In the Chinese mitten crab, the gene encoding transforming growth factor-β type I receptor (*EsTGFBRI*) is involved in muscle growth during molting [62]. In our study, *LvCHH Ia-2* knockdown upregulated TGF-β signaling pathway expression, which might contribute to body weight gain following frequent molting (Figure 7E). Moreover, in the dsCHH Ia-2 group, KEGG pathway analysis revealed that DEGs in thoracic ganglion were significantly enriched in the phosphatidylinositol signaling system (Figure 7E). Among these, 16 genes—including inositol 1,4,5-trisphosphate receptor (*IP_3_R*), protein kinase C (*PKC*), phospholipase C delta-4 (*PLC*)—were found to be downregulated, indicating that RNA interference led to suppression of the phosphatidylinositol signaling pathway. In both insects and crustaceans, neuropeptides typically exert their functions via G protein-coupled receptor (GPCR) pathways, resulting in the activation of signaling molecules such as cAMP, cGMP, or PLC. In the PLC-mediated cascade, PLC catalyzes the hydrolysis of PIP_2_ into diacylglycerol (DAG) and inositol trisphosphate (IP_3_), with the former activating PKC and the latter facilitating calcium ion release from the endoplasmic reticulum. [63,64] In neuronal cells, PKC activation is implicated in neurotransmitter release and synaptic plasticity. For instance, studies on diabetic rats with gastric motility disorders have shown that C-type natriuretic peptide (CNP) modulates the DAG–PKC signaling pathway by regulating PLC activity, thereby influencing the contractility of gastric smooth muscle [65]. Following the interference of *LvCHH Ia-2*, DEGs in the thoracic ganglia were found to be significantly enriched in neurotransmitter transport activity, with notable down-regulation of both *PLC* and *PKC* expression. These findings suggest that the PKC- and calcium ion-mediated neural signaling pathways may represent one of the mechanisms through which *LvCHH Ia-2* regulates physiological functions in shrimp, such as motor control and sensory signal transmission. For instance, during activities like swimming or foraging, calcium-mediated signaling in the thoracic ganglia conveys neural instructions from the brain to target muscle groups, enabling coordinated muscle contraction and relaxation. The RNAi-mediated silencing of *LvCHH Ia-2* resulted in a significant decrease in hemolymph glucose levels and the removal of molting inhibition. As a consequence of insufficient energy availability, it was likely that shrimp reduced locomotor activity by downregulating the aforementioned signaling pathways in order to minimize excessive energy expenditure. Although a certain degree of weight gain was observed in the dsCHH Ia-2 group, statistical analysis does not support the conclusion that this change was directly caused by gene interference. It is important to note that, during the experimental period, the shrimp underwent several molting phases. During these stages, some individuals were attacked and cannibalized by their specifics, influenced by behavioral factors, especially during the soft-shell period following molting, when their mobility was reduced, making them more susceptible to intraspecific predation. This phenomenon may have led to a reduction in the number of surviving individuals within the experimental group and could have introduced selection bias—specifically, the individuals that survived and contributed to the body weight data may have had greater competitive fitness or higher initial health status. As a result, the true physiological effects induced by gene interference may have been partially obscured. In conclusion, *LvCHH Ia* regulates growth and molting through multiple mechanisms, including direct metabolic control, indirect hormonal regulation, and adaptive responses to environmental changes.

In summary, *LvCHH Ia-2* appears to play an inhibitory role in the molting pathway, while *LvCHH Ia-1* promotes growth and development. The observed differences in body weight in this experiment are likely the result of multiple interacting factors. While *dsRNA* interference itself may have contributed to some phenotypic changes, uncertainties in the experimental process and the presence of external confounding factors prevented the weight changes in the *LvCHH Ia-2* group from achieving statistical significance. Future studies could improve the stability and reproducibility of the results by increasing the sample size, optimizing rearing conditions, and enhancing individual monitoring during critical periods.

## 4. Materials and Methods

### 4.1. Experimental Animals

The shrimp *L. vannamei* used in the experiment were sourced from Rizhao Hongqi Aquatic Products Co., Ltd. (Rizhao, China), and cultured under controlled laboratory conditions at the Coastal Aquaculture Laboratory of the Institute of Oceanology, Chinese Academy of Sciences (Qingdao, China). The culture system utilized sterilized seawater (3% salinity) processed through mechanical filtration and UV sterilization, with continuous aeration to maintain dissolved oxygen levels. Environmental parameters were regulated at 25 ± 1 °C (temperature), pH 7.5 ± 0.1, and a 12/12 h light–dark cycle throughout the experimental period. During the acclimatization period, half-volume water exchanges were performed daily using isosmotic seawater, while shrimp were fed ad libitum thrice daily (08:00, 12:00, 18:00) with a standardized commercial diet (Yantai Dale Feed Company, Yantai, China). All experimental procedures involving shrimp were conducted in accordance with the protocols sanctioned by the Animal Ethics Committee of the Institute of Oceanology, Chinese Academy of Sciences [2020(37)]. This study did not involve any rare or endangered species.

### 4.2. Identification and Analysis of CHH Gene Family Members

A total of 143 sequences were used as data sources for the analysis of the CHH gene family. These sequences included 53 CHH mRNA sequences extracted from *L. vannamei* genome and transcriptome data in our laboratory, and 90 additional sequences annotated as *L. vannamei* CHH retrieved from the NCBI database (https://www.ncbi.nlm.nih.gov/guide/proteins/, accessed on 9 October 2023). These sequences’ open reading frames (ORFs) were predicted using the NCBI ORF finder (https://www.ncbi.nlm.nih.gov/orffinder/, accessed on 15 October 2023) to determine the corresponding amino acid sequences. Subsequently, the conserved domains of the amino acid sequences were predicted using SMART (https://smart.embl.de/, accessed on 20 October 2023) and NCBI Batch CD-search (https://web.expasy.org/translate/, accessed on 22 October 2023). Based on these predictions, members of the CHH family containing complete CHH domains were identified. Finally, TBtools (version 2.153) was used to visualize the genomic localization of the CHH genes within the *L. vannamei* genome (http://www.shrimpbase.net/, accessed on 10 January 2024).

### 4.3. Characterization of CHH Gene Family Members

To classify the members of the CHH gene family in *L. vannamei*, we performed multiple sequence alignment using the Clustal W model in MEGA (version 11.0), followed by the construction of the CHH phylogenetic tree of *L. vannamei* using the neighbor-joining method. Meanwhile, 1000 bootstrap tests were conducted to assess the reliability of the tree topology, and the resulting phylogenetic tree was uploaded to the iTOL website (http://itol.embl.de/, accessed on 11 December 2024) for further visualization and annotation. Prediction of amino acid sequence motifs was performed using MEME (https://meme-suite.org/meme/tools/meme, accessed on 29 October 2023), and the identified motifs were compared with established functional sites in the Expasy prosite database (https://prosite.expasy.org/, accessed on 29 October 2023; version 2021_04). Promoter region sequences of CHH gene family members were extracted using TBtools (version 2.154), and transcription factor binding sites (TFBSs) were predicted using Animal TFDB3.0 (https://guolab.wchscu.cn/AnimalTFDB#!/, accessed on 20 November 2023) with a screening criterion of *p*-value < 10^−7^. The phylogenetic tree results, motif analysis, and TFBS prediction were integrated to infer the CHH family genes potentially involved in glucose metabolism in *L. vannamei*.

### 4.4. Gene Cloning

Based on the mRNA sequence of the key CHH gene (*LvCHH Ia*) for sugar metabolism obtained from the above screening process, upstream and downstream amplification primers were designed using Primer3 Plus (https://www.primer3plus.com/, accessed on 17 November 2023 ) and named LvCHH Ia-F/R (Appendix A). Subsequently, *L. vannamei* cDNA was used as a template for in vitro amplification, and the amplified products were verified by sequencing (Shanghai Sangon Biotech, Shanghai, China). The PCR product with correct sequencing results was purified using a PCR DNA purification kit (Takara Bio Inc., Kusatsu, Japan). Finally, the full-length sequence of the CHH gene was cloned into the pMD19-T vector (Takara Bio Inc., Kusatsu, Japan) in Escherichia coli and stored at −80 °C in 30% glycerol.

### 4.5. Gene and Protein Structure

MEGA 11.0 was used to align the sequences, and Jalview (version 2.11.4.1) was used to visualize the completed sequence alignment. SignalP-6.0 (https://services.healthtech.dtu.dk/services/SignalP-6.0/, accessed on 3 November 2023) was used to predict signal peptides, while GPS-ALL 2.0 was used to predict C-terminal lysine acetylation of LvCHH Ia-1 and LvCHH Ia-2 [66,67]. The AlphaFold Server (https://alphafoldserver.com/welcome, accessed on 13 February 2025 ) was used to predict the tertiary structure of proteins, and FTMap (https://ftmap.bu.edu/login.php?redir=/home.php, accessed on 15 February 2025) was used to simulate rigid docking on the protein surface with 16 small organic molecules probes. Fast Fourier transform (FFT) was used to predict interactions between amino acid residues and probe molecules within the three-dimensional structure [68,69]. The three-dimensional structures of the proteins were visualized using the PlayMolecule viewer (https://open.playmolecule.org/, accessed on 15 February 2025).

### 4.6. Gene Expression Pattern

The FPKM values of the CHH genes from the transcriptome data of different tissues of *L. vannamei*, previously published by our laboratory [70], were utilized as data sources. After log2 transformation and normalization, the processed data were imported into TBtools (version 2.154) to generate an expression heatmap, which illustrated the expression patterns of CHH gene family members across tissues. For the key CHH gene involved in glucose metabolism, we targeted two different splicing forms. Primers LvCHH Ia-1-qF/R and LvCHH Ia-2-qF/R (Appendix A) were designed using Primer3 Plus, and fluorescence quantitative PCR (qPCR) was performed to validate their expression distribution in 13 tissues.

### 4.7. RNA Interference

In this study, a dsRNA interference experiment was conducted to explore the regulatory role of the two alternative splicing forms of the *LvCHH Ia* gene of glucose metabolism and their effect on growth. Plasmid DNA was extracted from the reactivated *Escherichia coli* recombinant PMD19-T strain containing the *LvCHH Ia* gene using the EasyPure Plasmid Mini-Prep Kits (EM101-02, TransGen Biotech, Beijing, China). Using Primer3 Plus, dsRNA primers (LvCHH Ia-1-dsF/R and LvCHH Ia-2-dsF/R) containing T7 promoters were designed within the conserved domains of *LvCHH Ia-1* and *LvCHH Ia-2*. Enhanced green fluorescent protein (EGFP) was used as the control, and dsRNA primers EGFP-dsF/R were synthesized based on its sequence (Appendix A). The PCR product was purified using the Mini BEST DNA Fragment Purification Kit (Takara Bio Inc., Kusatsu, Japan), followed by in vitro transcription with the Transcript Aid T7 High Yield Transcription Kit (Thermo Fisher Scientific, Beijing, China). The dsRNAs of *EGFP*, *LvCHH Ia-1*, and *LvCHH Ia-2* were synthesized according to the manufacturer’s instructions (Thermo Fisher Scientific, Beijing, China). After the 1.5% agarose gel electrophoresis, the quality of the synthesized dsRNA was evaluated and stored at −80 °C.

A preliminary experiment was conducted to determine the optimal dosage of RNAi. Forty shrimp in the D1-D2 molting stage, with uniform sized (average weight 5.875 ± 0.6 g), were selected and divided into ten groups: including six experimental groups (three dsCHH Ia-1 groups and three dsCHH Ia-2 groups), three control groups (dsEGFP), and one PBS group, each containing four shrimp. Before injection, dsRNA concentrations were adjusted using 1×PBS. Except for the PBS group (injected with 10 µL of 1×PBS), all groups were injected with three gradient doses of dsRNA (0.2 µg, 1 µg, and 2 µg per individual), with a total injection volume of 10 µL per shrimp. After 48 h, hemolymph, thoracic ganglion, and hepatopancreas tissues were collected from each shrimp. Hemolymph was mixed with ice-cold, sugar-free anticoagulant (1:1 ratio) and centrifuged at 4 °C for 10 min at 800× *g*. Hemolymph glucose levels were measured using an O-toluidine glucose detection kit (Beyotime, Shanghai, China). RNAi effectiveness was assessed via RT-qPCR, and the optimal injection doses of dsRNA for the dsCHH Ia-1 and dsCHH Ia-2 groups were determined.

The shrimp in the D1-D2 molting stage with an average body weight of 3.2 ± 0.6 g and a body length of 6.7 ± 0.7 cm were used for formal RNAi experiment. Three control groups (uninjected, PBS, dsEGFP) and two treatment groups (dsCHH Ia-1 and dsCHH Ia-2) were established. The uninjected group served as an additional control, while the remaining four groups were set up with three parallel replicates each, totaling 13 groups. Each group consisted of 30 shrimp, totaling 390 shrimp. The optimal RNAi dose was administered based on the shrimp’s body weight ratio. The experiment lasted 23 days, with one injection every five days, totaling five dsRNA injections. Tissue samples, including hepatopancreas and thoracic ganglia, were collected 48 h post-final injection. Four biological replicates were set from each group, each replicate comprising four shrimp pooled into a single sample. The daily number of molts in each group was documented during the study period, and shrimp weight and length measurements were obtained prior to sampling. Immediately after collection, the tissue samples were flash-frozen in liquid nitrogen and stored at −80 °C until further analysis. For gene expression analysis, 18S rRNA was used as the internal reference, and the relative expression levels of *LvCHH Ia-1* and *LvCHH Ia-2* in both the experimental and control groups were calculated using the 2^−∆∆*C*t^ method. Each sample included four technical replicates, and all experiments were performed according to the instructions provided with the SuperReal PreMix Plus (SYBR Green) kit (Tiangen, Beijing, China). Levels of cAMP and cGMP were measured using the cAMP ELISA kit and cGMP ELISA kit (Lengton Bioscience, Shanghai, China), respectively.

To explore the effects of *LvCHH Ia-1* and *LvCHH Ia-2* knockdown on gene expression changes during metabolic processes and growth in *L. vannamei*, we performed transcriptome analyses on two key tissues: the highly expressed thoracic ganglia and metabolism-associated hepatopancreas.

### 4.8. Recombinant Protein Injection

In order to verify the effect of the *LvCHH Ia* gene on glucose metabolism, a recombinant protein injection overexpression experiment was conducted. The mature peptide segments of *LvCHH Ia-1* and *LvCHH Ia-2* were cloned into the PET-32a vector (Beijing TransGen Biotech, Beijing, China) to construct recombinant expression plasmids. Subsequently, the two recombinant LvCHH-PET-32a-His plasmids were transformed into DE3 (BL21) competent cells (TransGen Biotech, Beijing, China). The bacterial solution was inoculated into liquid LB medium containing ampicillin at a 1:100 dilution ratio and activated overnight activation at 37 °C. The activated bacterial culture was re-inoculated and grown until the OD600 reached 0.4–0.6. Isopropyl β-d-thiogalactoside (IPTG) was added to the culture medium at a final concentration of 0.5 mMol/L to induce protein expression. The culture was maintained at 16 °C and 180 rpm for 22 h. After harvesting, the bacterial cells were resuspended in a phosphate buffer and disrupted using an ultrasonic instrument in an ice-water mixture for 30 min. The cell lysate was centrifuged at 8000 rpm for 30 min at 4 °C to isolate the soluble recombinant protein. The fusion protein in the lysate was identified via 12.5% sodium dodecyl sulfate-polyacrylamide gel electrophoresis (SDS-PAGE) and purified using cobalt ion affinity resin (TALON Metal Affinity Resin, Takara Bio, Kyoto, Japan). The purified fusion protein was dialyzed against Tris-HCL buffer and treated with high-specificity recombinant enterokinase (Vazyme Biotech, Nanjing, China) to remove the His and lytic tags. The resulting CHH protein was dialyzed into PBS buffer, verified by SDS-PAGE, concentrated using ultrafiltration tubes, and stored at −80 °C for subsequent applications.

To determine the optimal protein injection dosage for the overexpression experiment, 120 uniformly sized shrimp in the D1–D2 molting stage were selected for a pre-experiment. The average weight of the shrimp was recorded as 6.27 ± 0.1 g, and their body length was measured at 8.19 ± 0.1 cm. In the pre-experiment, the 120 shrimp were divided into four groups: uninjected group, PBS-injected group, rLvCHH Ia-1 group, and rLvCHH Ia-2 group. The latter two groups received injections of three distinct protein dosage (10 poml, 100 poml, 200 poml), totaling eight experimental groups. Prior to the recombinant protein injection, the experimental shrimp were subjected to a 12 h starvation period. Three distinct sampling time points (1 h, 3 h, and 6 h) were established for each group. Hemolymph samples were collected from five shrimp in each group at the three different time intervals following protein injection. The procedures for hemolymph collection and glucose testing were identical to those used in the RNAi pre-experiment. The optimal protein injection dosage for the rLvCHH Ia-1 and rLvCHH Ia-2 groups was determined by analyzing hemolymph glucose concentrations.

In the formal recombinant protein injection experiment, shrimp with an average body weight of 6.44 ± 0.1 g and an average body length of 8.44 ± 0.2 cm were used. A total of 180 healthy shrimp in the D1~D2 molting stage were selected for in vivo injection experiments and divided into four groups: uninjected control, PBS-injected, rLvCHH Ia-1, and rCHH- Ia-2 groups. Excluding the uninjected group, the remaining three groups were further subdivided into three treatment subgroups based on varying sampling intervals (1 h, 3 h, and 6 h post protein injection), resulting in 10 experimental groups, with 18 shrimp per group. Prior to the experiment, the shrimp underwent a 12 h starvation period and were subsequently injected with the appropriate concentration of recombinant proteins determined in the pre-experiment. Hemolymph samples were collected from each shrimp individually, while the remaining tissues were collected and pooled in four biological replicates per group, with each replicate consisting of tissue samples from three shrimp combined into a single sample. The glucose content in the hemolymph was measured after collecting the hemolymph supernatant. The tissue processing methods and hemolymph glucose concentrations measurements were consistent with those used in the RNAi experiment.

### 4.9. Statistical Analysis

In the RNAi experiments, the treatment and control groups’ relative expression levels were analyzed using a *t*-test. This statistical method was also employed for analyzing data on growth traits and molting counts. A one-way ANOVA was used to analyze the hemolymph glucose and cAMP/cGMP assay data. In cases where significant differences existed between groups, Tukey’s post hoc test was conducted for multiple comparisons between means. Significant differences between different treatments are shown as * *p*-value < 0.05, ** *p*-value < 0.01, **** *p*-value < 0.0001. All statistical analyses were conducted using GraphPad Prism (version 10.1.2).

## 5. Conclusions

In this study, a comprehensive investigation was conducted on the CHH family member *LvCHH Ia* in *L. vannamei*, which is associated with glucose metabolism. Through the integration of genomic, transcriptomic, RNA interference, and overexpression approaches, two alternative splicing variants of *LvCHH Ia* were identified. These variants appear to regulate hemolymph glucose homeostasis by simultaneously promoting gluconeogenesis and downregulating glycolysis pathways. Among them, the short splicing variant *LvCHH Ia-1* exhibited a more sustained effect on glucose regulation compared to the longer variant *LvCHH Ia-2*. Beyond glucose metabolism, *LvCHH Ia* may also participate in other physiological processes such as growth and molting regulation. Notably, *LvCHH Ia-2* is likely to exert a stronger role in molting control, potentially through its involvement in chitin metabolic pathways.

## Figures and Tables

**Figure 1 ijms-26-04612-f001:**
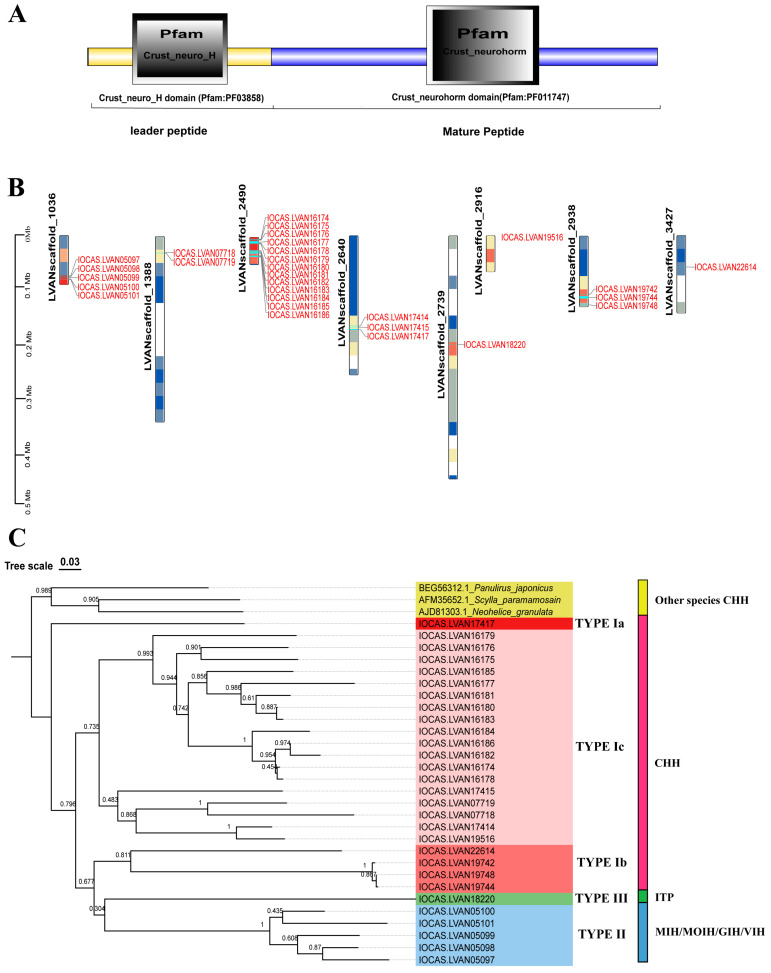
Gene structure, chromosome localization, and evolutionary analysis of the CHH gene family in *L. vannamei*. (**A**) Distinctive domain of the CHH genes; (**B**) Chromosomal localization of the CHH genes; (**C**) Phylogenetic tree and classification of CHH family genes in *L. vannamei*.

**Figure 2 ijms-26-04612-f002:**
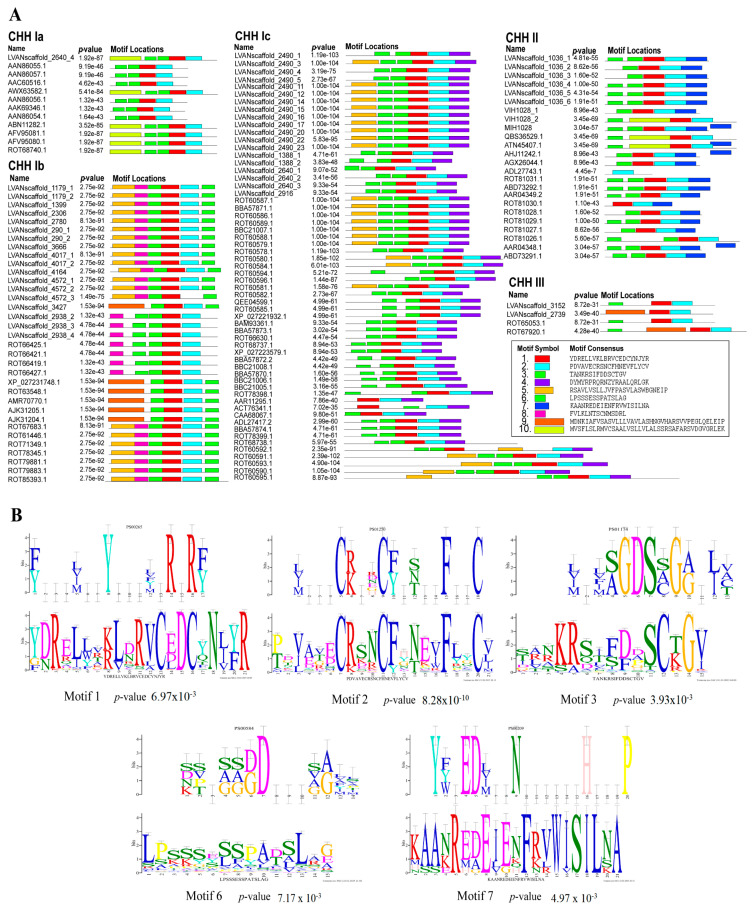
Motifs of the CHH sequences of *L. vannamei*. (**A**) CHH family protein motif prediction; (**B**) Sequence logo and alignment analysis of CHH protein motif functional sites. The lower figures take the predicted CHH protein motifs as the query motif. The upper figure shows the target motifs that were compared.

**Figure 3 ijms-26-04612-f003:**
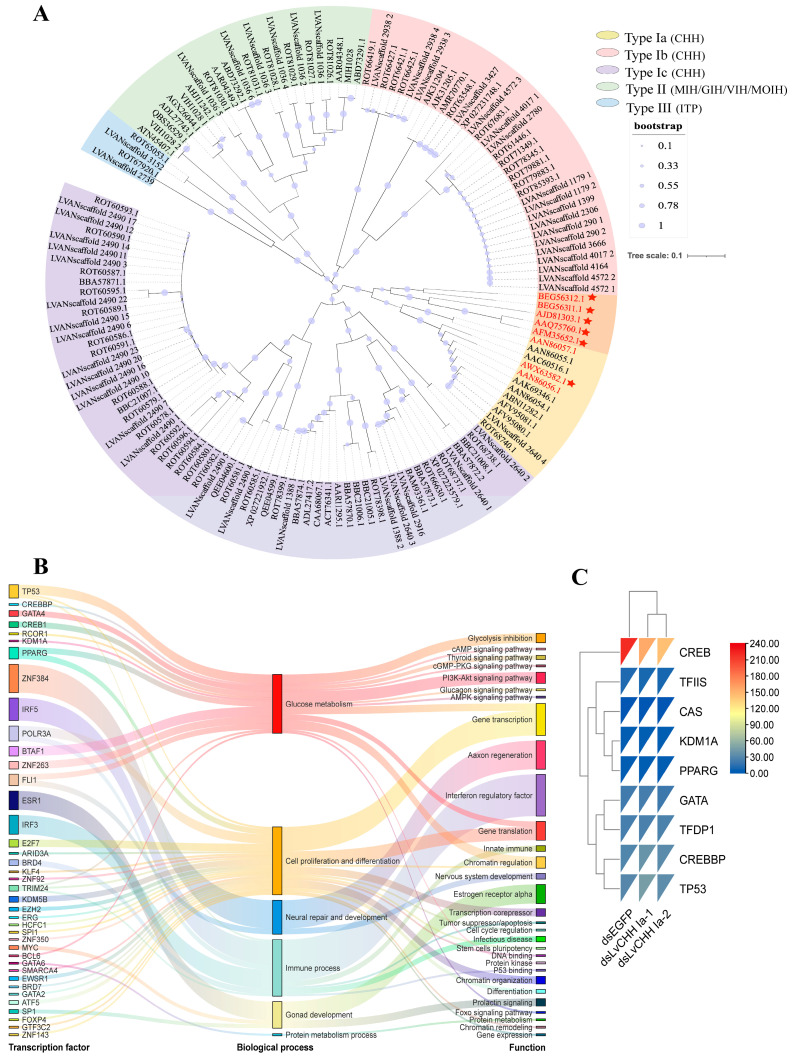
Identification and functional characterization of key CHH genes involved in glucose metabolism. (**A**) Phylogenetic tree of CHHs. Genes highlighted in red are the reported CHH genes for glucose metabolism in crustaceans. The orange blocks and red stars represent CHH proteins associated with sugar metabolism in other crustaceans; (**B**) Transcription factor binding sites of the *CHH Ia* gene; (**C**) Expression comparison of the predicted transcription factors and random background genes (*TFDP1*, *TFIIS*, and *CAS*) in the RNA interference (RNAi) transcriptome of the *LvCHH Ia* gene.

**Figure 4 ijms-26-04612-f004:**
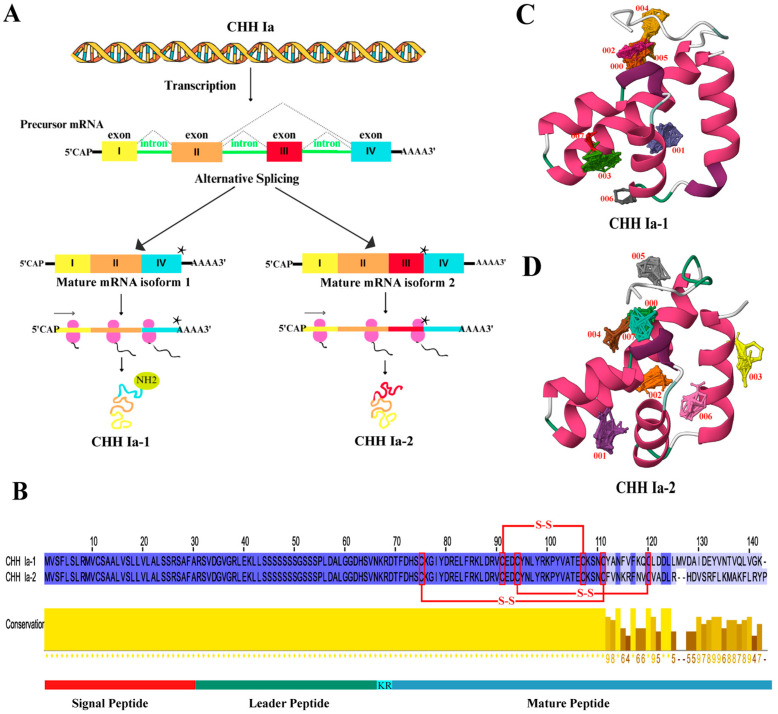
Analysis of *LvCHH Ia* gene and its structure. (**A**) *LvCHH Ia* alternative splicing mechanism, the star represents a stop codon; (**B**) Amino acid sequence alignment of LvCHH Ia-1 and LvCHH Ia-2; (**C**) Tertiary structure and potential binding hotspot prediction of *LvCHH Ia-1*; (**D**) Tertiary structure and potential binding hotspot prediction of LvCHH Ia-2. α-helix (pink), irregular coil (green and white).

**Figure 5 ijms-26-04612-f005:**
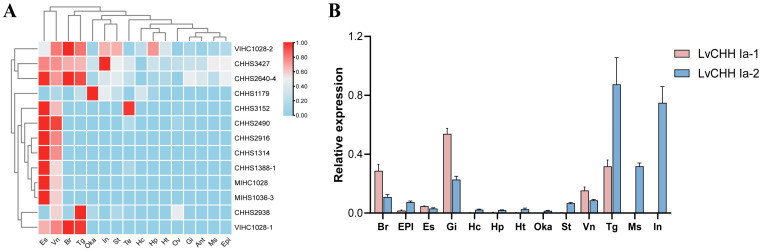
Tissue expression patterns of CHH genes. (**A**) Comparison of the expression levels of CHH gene family members in different tissues; (**B**) Relative expression levels of *LvCHH Ia-1* and *LvCHH Ia-2* in different tissues.

**Figure 6 ijms-26-04612-f006:**
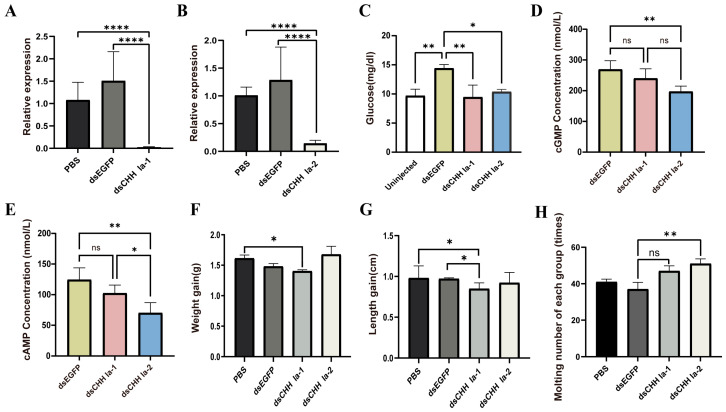
Effects of *LvCHH Ia* knockdown on hemolymph glucose, growth, and molting in *L. vannamei*. (**A**) Efficiency of the *LvCHH Ia-1* RNAi; (**B**) Efficiency of the *LvCHH Ia-2* RNAi; (**C**) Alterations in hemolymph glucose levels 48 h post-*LvCHH Ia* knockdown; (**D**) Changes in hemolymph cAMP levels following RNAi; (**E**) Changes in hemolymph cGMP levels following RNAi; (**F**) Variations in body weight after RNAi; (**G**) Variations in body length after RNAi; (**H**) Molting frequency following RNAi. Significant differences between different groups were shown as * *p*-value < 0.05, ** *p*-value < 0.01, **** *p*-value < 0.0001.

**Figure 7 ijms-26-04612-f007:**
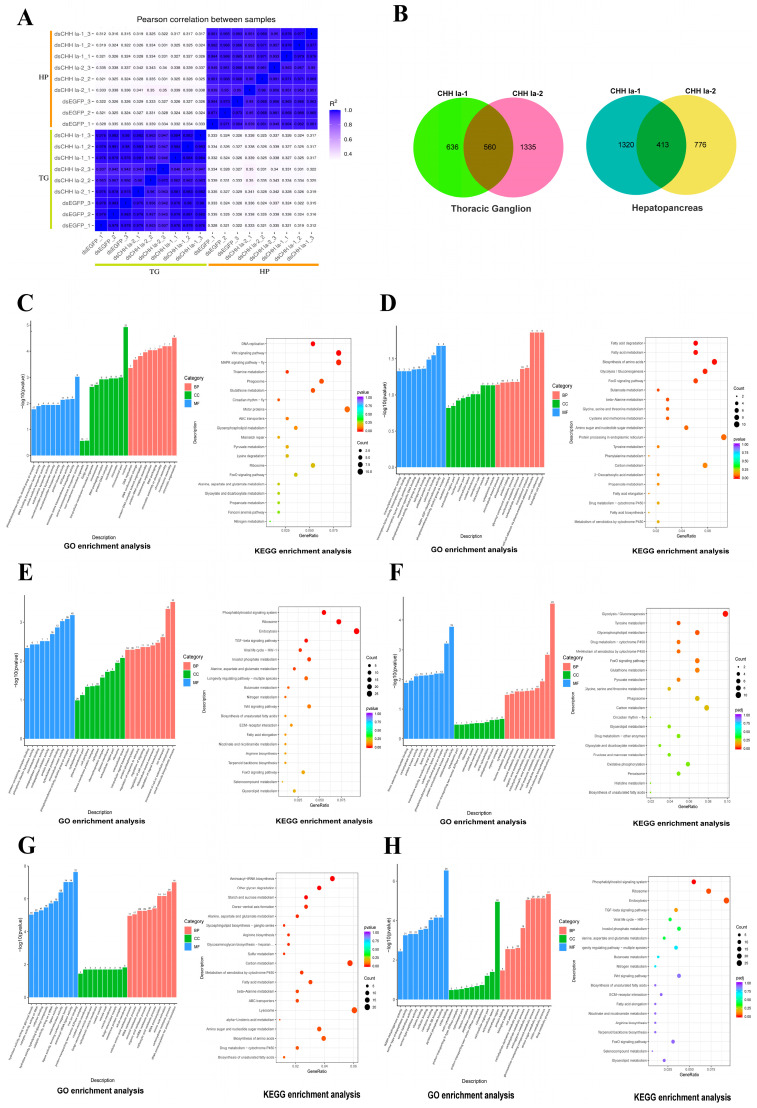
Transcriptome analysis reveals gene expression changes after *LvCHH Ia* knockdown. (**A**) Square of Pearson correlation coefficient (R2) between samples; (**B**) Number of DEGs in hepatopancreas and thoracic ganglia between dsCHH Ia-1 group and dsCHH Ia-2 group; (**C**) Top 30 GO terms and top 20 KEGG pathways enriched by the common DEGs in the thoracic ganglion from both dsCHH Ia-1 and dsCHH Ia-2 groups; (**D**) Top 30 GO terms and top 20 KEGG pathways enriched by the unique DEGs in the thoracic ganglion from the dsCHH Ia-1 group; (**E**) Top 30 GO terms and top 20 KEGG pathways enriched by the unique DEGs in the thoracic ganglion from the dsCHH Ia-2 group; (**F**) Top 30 GO terms and top 20 KEGG pathways enriched by the common DEGs in the hepatopancreas from both dsCHH Ia-1 and dsCHH Ia-2 groups; (**G**) Top 30 GO terms and top 20 KEGG pathways enriched by the unique DEGs in the hepatopancreas from the dsCHH Ia-1 group; (**H**) Top 30 GO terms and top 20 KEGG pathways enriched by the unique DEGs in the hepatopancreas from the dsCHH Ia-2 group.

**Figure 8 ijms-26-04612-f008:**
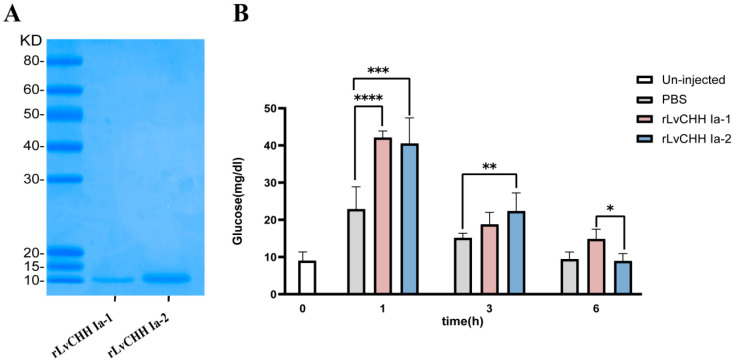
Recombinant LvCHH Ia and its injection experiment. (**A**) Purified mature peptides of rLvCHH Ia-1 and rLvCHH Ia-2; (**B**) Glucose content in hemolymph at different time points following rLvCHH Ia-1 and rLvCHH Ia-2 injection. Significant differences between different groups were shown as * *p*-value < 0.05, ** *p*-value < 0.01, *** *p*-value < 0.001, **** *p*-value < 0.0001.

## Data Availability

All data have been included in the manuscript.

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
