# Peer review of "Identification and Functional Analysis of a Key Gene in the CHH Gene Family for Glucose Metabolism in the Pacific White Shrimp Litopenaeus vannamei"

_ijms, 2025, doi:10.3390/ijms26104612_

Round 1

Reviewer 1 Report

Comments and Suggestions for Authors

Author Response

Dear Editor and Reviewers,

We would like to sincerely thank you for your thoughtful and constructive feedback on our manuscript. We are very grateful for the time and effort you invested in reviewing our work, as well as for the insightful comments and suggestions that have helped us to improve the clarity, rigor, and overall quality of the manuscript.

We have carefully addressed each of the reviewers’ comments and revised the manuscript accordingly. In the following pages, we provide a detailed, point-by-point response to each comment, along with a description of the corresponding revisions made to the manuscript. All changes in the revised version are highlighted in red for your convenience.

Comments 1:The entire Abstract requires rewriting. Firstly, the identification of CHH family genes constitutes a significant component of the article, but it was merely brushed over in the abstract. Notably, the authors employed three figures (out of a total of eight) to illustrate this process. Secondly, the authors should provide some key results in the abstract to support their conclusion. In fact, some of their conclusions lack sufficient data to support them, and I will specifically discuss these issues in the subsequent comments.

Response 1:

Thank you for your comment. We have rewritten the abstract, adding a description of the CHH family gene identification. At the same time, we have also included some key results to support the conclusions of each part. Please refer to the new abstract for details.

Comments 2:Since CHH Ia-1 and CHH1a-2 are produced by variable splicing, it is challenging to correctly distinguish their expression by qPCR. For example, if the primer is designed in the region of exon III, then there is no doubt that CHH Ia-2 can be specifically amplified; however, if the primer is designed in any region of I, II, or IV, it will amplify both CHH Ia-1 and CHHIa-2 products. It is recommended that the authors highlight in the supplemental material the range of sites and products included in the qPCR primer design (I was unable to verify this as I was unable to open the data3 file in the supplemental material).

Response 2:

Thank you for your valuable suggestion. We fully agree with your assessment of the challenges associated with distinguishing their expression using qPCR, and we appreciate you bringing this important point to our attention.

After careful consideration, we've found that the primers we initially used for amplifying LvCHH Ia-1 during qPCR amplification indeed amplified both LvCHH Ia-1 and LvCHH Ia-2 simultaneously. It's clear that the description of the tissue expression distribution of LvCHH Ia-1 in the article is incorrect. To address this concern, new qPCR primers for LvCHH Ia-1 were designed, and its expression pattern was revalidated through qPCR using the same batch of whole-tissue samples from Litopenaeus vannamei. The primer sequence of LvCHH Ia-1 has been revised in Supplementary S3. To address this issue, we redesigned new qPCR primers for LvCHH Ia-1 of Litopenaeus vannamei and re-validated its expression pattern using the same batch of tissue samples through qPCR. The new LvCHH Ia-1 primer (LvCHH Ia-1-F/R) covers the junction of exon II and exon IV, which can specifically amplify LvCHH Ia-1 in mRNA without including LvCHH Ia-2. The revised primer sequences are provided in Supplementary Material S3. We used different colours to distinguish the different exon ranges of the mature mRNA sequences of LvCHH Ia-1 and LvCHH Ia-2. At the same time, the primer sites of the qPCR experiment have been bolded, and the product ranges have been underlined. 

Additionally, transcriptomic read count and TPM data from various tissues of Litopenaeus vannamei ([70] Zhang et al., 2019) were double-checked to verify the findings. I In the heatmap, color blocks represent the logâ‚‚-transformed and normalized count values of the two alternative splicing isoforms across different tissues, reflecting the comparative expression of these isoforms in various tissues. It can be noted that discrepancies in gene expression may arise due to differences in batch, size, and culture environment of the shrimp used in the qPCR and transcriptome datasets. Nevertheless, in both validation approaches, the CHH gene consistently showed high expression in the thoracic ganglion, gills, abdominal ganglion, and brain. These results indicate that the heatmap data are generally consistent with the qPCR findings.

In accordance with the new results, corresponding revisions have been made in Results and discussion of the manuscript. In addition, the supplementary materials have been re-uploaded. We sincerely apologize for any inconvenience this may have caused and would like to once again express our sincere gratitude for your valuable comments.

Comments 3: Figure 4A needs to be modified: the “NH2” end is not shown in CHH Ia2; the “*” may imply a termination codon, which is not shown in CHH Ia1; the precursor mRNA for CHH Ia1 should be changed to Mature mRNA isoform 1 instead of Mature mRNA isoform 2; in the corresponding text, the KR site should be (67-68) instead of (69, 69).

Response 3:

Thank you for your meticulous review and constructive suggestions about Figure 4A. We sincerely appreciate your attention to these details, which have greatly helped us improve the accuracy of our manuscript.

The original intention of "NH2" was to represent amidation modification. This post-translational modification was only predicted in the deduced LvCHH Ia-1 protein, but not in LvCHH Ia-2. Therefore, we did not mark it in LvCHH Ia-2.

As for the " * " indicating a termination codon, we have presented it after exon IV of LvCHH Ia-1 in Figure 4A. We also corrected the mislabeling of the mature mRNA isoform names in the figure, which was causing confusion. Additionally, in the corresponding text of Figure 4, We have carefully checked the sequence data and have revised the KR site from (69, 69) to (67 - 68).

Comments 4: In Figure 5B, the presentation of error bars is not uniform.

Response 4:

We agree with your opinion. The inconsistent presentation of error bars in Figure 5B indeed reflects our negligence and lack of rigor in the process of chart making. As an important element for demonstrating the dispersion of data and the reliability of experiments, the way error bars are presented should be kept uniform and standardized to ensure that readers can accurately understand the data information. We have a profound understanding of the seriousness of this issue and would like to express our apologies to you again. The issue has been given careful attention in the updated qPCR result figure, and appropriate modifications have been implemented.

Comments 5: Figure 6 needs to be revised: There are problems with the units of the vertical axis in Figures 6D and 6E; Figure 6F does not indicate significance because the text claims that "the weight gain of the dsCHH Ia-2 group exceeded that of the control group"; the content pointed to by the asterisk in Figure 6G is unclear; the number of molting times (times) in Figure H is confusing, especially the corresponding values: Did this shrimp molt 40 times during the 26-day experiment? Or did the author count the number of shrimp molting in each group during the experimental period? Then the name of the vertical axis should be modified. 

Response 5:

Thank you for your suggestion. We have conducted a comprehensive review and revision of the relevant content. The specific responses and improvement modify in the “Result” section 2.5 (page 10) Figure 6 are reported a s follows:

  1. Upon verification, the units on the vertical axes of the Figure 6D and Figure 6E are marked incorrectly. We have corrected the error unit markings on the vertical axes in Figures 6D and 6E, and the corrected content is "(nmol/L)".

The statement that "the weight gain of the dsCHH Ia-2 group exceeded that of the control group" is incorrect and does not match the content of the graph bars. In the paper, I have re-described it and modified it to “ In contrast, the dsCHH Ia-2 group exhibited greater weight gain compared to the PBS and dsEGFP groups (Figure 6F), whereas their increase in body length was slightly lower than that observed in the other two groups (Figure 6G). Although a trend to-ward enhanced weight gain was observed in the LvCHH Ia-2 dsRNA interference group by the end of the experiment, statistical analysis revealed that the difference was not statistically significant (p-value > 0.05)” .

Additionally, in the section 3.4 (page 18) of the “discussion” relevant content has been added to “Although a certain degree of weight gain was observed in the dsCHH Ia-2 group, statistical analysis does not support the conclusion that this change was directly caused by gene interference. It is important to note that, during the experimental period, the shrimp underwent several molting phases. During these stages, some individuals were attacked and cannibalized by their specifics, influenced by behavioral factors, especially during the soft-shell period following molting, when their mobility was reduced, making them more susceptible to intraspecific predation. This phenomenon may have led to a reduction in the number of surviving individuals within the experimental group and could have introduced selection bias specifically, the individuals that survived and contributed to the body weight data may have had greater competitive fit-ness or higher initial health status. As a result, the true physiological effects induced by gene interference may have been partially obscured. In conclusion, LvCHH Ia regulates growth and molting through multiple mechanisms, including direct metabolic control, indirect hormonal regulation, and adaptive responses to environmental changes. In summary, LvCHH Ia-2 appears to play an inhibitory role in the molting pathway, while LvCHH Ia-1 promotes growth and development. The observed differences in body weight in this experiment are likely the result of multiple interacting factors. While dsRNA interference itself may have contributed to some phenotypic changes, uncertainties in the experimental process and the presence of external confounding factors prevented the weight changes in the LvCHH Ia-2 group from achieving statistical significance. Future studies could improve the stability and reproducibility of the results by increasing the sample size, optimizing rearing conditions, and enhancing individual monitoring during critical periods.”

  1. In Figure 6G, the "*" indicates that there are significant differences in the change of body length between the dsCHH Ia-1 group and both the PBS and dsEGFP groups. We have made modifications in Figure 6G to make the expression clearer.
  2. The "molting number" in Figure 6H refers to the total number of molting events in each group during the experiment. To make the meaning clearer, we have modified the corresponding content in the paper to "By recording the molting numbers of shrimp in each group during the experiment, we found that injecting two types of dsCHH Ia increased shrimp molting frequency (Figure 6H)". In addition, the name of the vertical axis in Figure 6H has been modified to "Molting number of each group(times)".
  3. The modified Figure6 is as shown below.

Comments 6: The results corresponding to Figure 6 also raise another question: why did the authors measure the levels of cAMP and cGMP in hemolymph? cAMP and cGMP are intracellular second messengers, and the authors should have measured their levels in the target tissues. Given that the decline in cAMP levels is the only result that led the authors to conclude that the two CHH variants might regulate gluconeogenesis through the cAMP-PKA pathway, they should have measured the levels of cAMP and cGMP in the hepatopancreas. Therefore, as mentioned earlier, this conclusion is not appropriate to be included in the abstract because it lacks key data support.

Response 6:

 Thank you for your careful review of the manuscript and the valuable suggestions. Your doubts about measuring the levels of cAMP and cGMP in hemolymphs are very professional and to the point. We fully agree with this. In the initial research design, we chose to detect the level of second messengers in hemolymph, mainly based on the following considerations: Hemolymph, as a circulating fluid similar to blood in crustaceans, can reflect the dynamic balance of signal molecules in the overall body and initially provide us with clues about the changes in the second messenger system. These fluctuations reflect a generalized endocrine response rather than tissue-specific intracellular signaling events. Accordingly, we acknowledge that our original interpretation of hemolymph cAMP/cGMP levels as indicative of glucose metabolism regulation in the hepatopancreas may not have been precise, we have revised the corresponding statements in the manuscript to reflect a more accurate understanding. Regarding this part of the experiment, we will refine and modify the test plan in the later stage and conduct the test again.

Comments 7: Transcriptome analysis: In my opinion, the presentation of Figure 7 is not necessary and could be considered for inclusion in the supplementary materials. Instead, since Figure 7B shows the intersection of differentially expressed genes between the two treatments, the authors should focus on presenting the GO and KEGG terms of these common differentially expressed genes, as well as those of the non-intersecting differentially expressed genes, to more clearly demonstrate the functional differences between the two CHH variants.

Response 7:

We sincerely appreciate your constructive comments and valuable suggestions. In response to your concern, enrichment analyses were re-conducted separately for the commonly and differentially expressed genes identified in both the dsCHH Ia-1 and dsCHH Ia-2 treatment groups. This revised approach has indeed allowed for a more intuitive and functionally meaningful distinction between their roles in the thoracic ganglion and hepatopancreas. It is worth noting that the conclusions derived from this separate analysis remain largely consistent with those obtained using the previous method. The results and corresponding discussion have been updated in the manuscript to reflect this refined interpretation.

Comments 8:Figure 8, the claim that “The capacity of rLvCHH Ia-1 to rapidly elevate hemolymph glucose levels within one hour exceeded that of rLvCHH Ia-2” is nor correct. No significantly difference was observed between the two injection groups. Nevertheless, it is safe to say “rLvCHH Ia-1 exhibited a more prolonged hyperglycemic effect.”

Response 8:

We accept your professional opinion. The statement in the original text that “The capacity of rLvCHH Ia-1 to rapidly elevate hemolymph glucose levels within one hour exceeded that of rLvCHH Ia-2” is indeed misleading. Although we obtained the trend differences by analyzing the experimental data, after re-checking and statistically analyzing, the difference between the two groups at the 1-hour time point did not reach the significant difference level (p > 0.05). I sincerely apologize to you here. We have made the following revisions to the original text:

In the "Results" section2.5, the original sentence was modified as follows: " It is worth noting that the rLvCHH Ia-1 exhibited a more prolonged hyperglycemic effect.” In the " Discussion" section3.2, the original sentence was modified as follows: "Notably, the short splice variant LvCHH Ia-1 exhibits a more sustained glucose increasing effect than the long splice variant LvCHH Ia-2.”

Comments 9: Section 3.2, “This epigenetic change improves protein stability and signaling pathway activation, ……” The C-terminal amidation is not an epigenetic change, it is a posttranslational modification.

Response 9:

Thank you for pointing out this mistake. Epigenetic modification mainly involves the regulation of heritable gene expression such as DNA methylation and histone modification that do not change the DNA sequence, while post-translational modification is a covalent modification of already synthesized proteins. There is an essential difference between of them.

Comments 10: Section 3.3, my greatest concern about this part is that the phosphatidylinositol pathway is only enriched in the thoracic ganglion in CHH Ia-2 disturbing group, but not in the hepatopancreas. The authors should limit their discussion of its significance to the nervous system and not consider it a universal phenomenon. This is also the reason why I suggest that the authors discuss the common and different enriched pathways separately. Secondly, the author has some confusion regarding some basic concepts. Neuropeptides including CHH usually exert their effects through the GPCR pathway, which typically leads to an increase in cAMP, cGMP or PLCβ. In the PLCβ-mediated pathway, PLCβ catalyzes PIP2 to generate DAG and IP3, with the former activating PKC and the latter activating the endoplasmic reticulum calcium ion channel. The other pathway mentioned by the author, where PI3K phosphorylates PIP2 to PIP3, involves the activation of PI3K and belongs to the tyrosine kinase receptor-mediated cell signaling pathway. Since the author only mentioned the downregulation of PKC and the so-called PIP-PDE (which I think might be PLC), this can only indicate a weakening of GPCR signaling. As for the activation of PI3K (if it exists), the author should refer to the tyrosine kinase receptor-mediated signaling pathway rather than GPCR. One possibility is related to the insulin pathway, which can activate the PI3K-Akt pathway and further act on TOR and Foxo, etc. These pathways are all mentioned in the manuscript. In fact, the decrease in CHH would lead to a reduction in blood glucose levels, thereby weakening the insulin pathway signal. This indirect regulatory effect should not be ignored. Therefore, the author can refer to this suggestion to reorganize this part of the discussion.

Response 10:

We have made revisions in the text. Additionally, we have revised our discussion of the phosphatidylinositol signaling pathway to clarify that its enrichment was observed specifically in the thoracic ganglion of the dsCHH Ia-2 group and not in the hepatopancreas. We have adjusted the language accordingly to avoid overstating its generality and to emphasize its relevance to the nervous system.

Comments 11: Please check the entire text to ensure that all gene names are in italics. As there are numerous results in the article, it is recommended to indicate the locations of the relevant charts and graphs that support their conclusions in discussion section. Additionally, some of the author's statements lack literature citations. For instance, in the introduction, they state "Members of the CHH family exhibit significant sequence conservation...", which requires a corresponding reference here; further on, "Amino acid mutations at certain positions can lead to partial or complete loss of CHH's hyperglycemic activity", if this statement is not from reference 26, a corresponding reference should be provided. However, if it is from reference 26, it is recommended to delete this sentence as its meaning is repetitive of the previous description.

Response 11:

Thank you very much for your suggestion.

Check on the italic format of gene names: We have carried out a comprehensive and meticulous inspection of the entire manuscript to ensure that all gene names are presented in italic format. To avoid any omissions, we utilized the find-and-replace function in the document as an aid for the inspection, and also conducted multiple rounds of manual rechecks to guarantee the standardization and uniformity of the format.​

Addition of chart references in the discussion section: In the discussion section, we have clearly indicated the locations of the relevant charts and graphs that support the conclusions for each result. By adding specific chart numbers, readers can more intuitively link the textual discussions with the data charts and graphs, enhancing the logicality and readability of this manuscript.

Supplementation and optimization of literature citations: Regarding the statement "Members of the CHH family exhibit significant sequence conservation..." in the introduction, we have supplemented the relevant reference [26]. This reference collected genomic and transcriptomic data of various crustacean species from public databases and published literature, and elaborated on the research findings regarding the sequence conservation of CHH family genes, enhancing the reliability of this statement. Concerning the sentence "Amino acid mutations at certain positions can lead to partial or complete loss of CHH's hyperglycemic activity", upon verification, it was found that this content did indeed overlap with original reference 26. We have deleted this sentence to ensure that the discussion is concise and clear.

Supplementary references:

Chang, W. H., and A. G. Lai. "Comparative Genomic Analysis of Crustacean Hyperglycemic Hormone (CHH) Neuropeptide Genes across Diverse Crustacean Species." F1000Res 7 (2018): 100.

We deeply appreciate your contributions and hope that the revised manuscript meets your expectations.

With best regards, 

The Authors 

Reviewer 2 Report

Comments and Suggestions for Authors

The manuscript by Zhang et al. studied the role of crustacean hyperglycemic hormone (CHH) gene family members in the regulation of glucose metabolism in Pacific white shrimps. In brief, they conducted bioinformatics analysis and identified two alternative splicing isoforms (LvCHH Ia-1 and -2). They carried out RNAi experiments and recombinant protein injection to knock down and overexpress these two splicing variants. They observed their distinct functions in modulating gluconeogenesis/glycolysis pathways. 

Comments and suggestions:
1.    Page 5, top, “A prediction motif analysis was performed…” It sounds confusing for what is being predicted. 
2.    Page 5, Figure 2, more details are expected for the legend. For example, what does the upper and lower panel represent in 2B? What does the error bar represent? And the legend of 2B (“CHH motif functional sites”) is too abstract and doesn’t reflect the figure (sequence logo representation). 
3.    Page 6, top, “To further investigate the classification and functional characterization…” The phylogenetic analysis is just inference. It doesn’t provide any “functional” evidence. The language “functional characterization” is overloaded. 
4.    Page 6, “To verify whether CHH Ia is a key gene…” Here, the authors only showed the TFBS of CHH Ia genes are likely to be enriched in glucose metabolism pathways, but no quantitative analysis. As this study does not gather evidence that these TFBS are indeed open chromatin and bound by glucose metabolism transcription factors, it is not sufficient to claim that CHH Ia genes are under regulatory control of glucose metabolism. Therefore, the authors are recommended to make a comparison to a background scenario where random genes are selected. 
5.    Figure 5A, how is the “relative expression” quantified? Is it from PCR or RNA-seq? The Methods (Section 4.6) suggest it was “FPKM values” but not very clear here. 
6.    Page 10, bottom, “All raw reads were deposited at the NCBI Sequential Read Archive (SRA) (PRJNA1240581)” This dataset cannot be found by the said accession number (https://www.ncbi.nlm.nih.gov/search/all/?term=PRJNA1240581). Could the authors double check? 
7.    Figure 7C-F, the barplots have illegible x-labels. The authors are expected to improve the figure resolution.   

Author Response

 Dear Reviewer,

We would like to sincerely thank you for your thoughtful and constructive feedback on our manuscript. We are very grateful for the time and effort you invested in reviewing our work, as well as for the insightful comments and suggestions that have helped us to improve the clarity, rigor, and overall quality of the manuscript.

We have carefully addressed each of the reviewers’ comments and revised the manuscript accordingly. In the following pages, we provide a detailed, point-by-point response to each comment, along with a description of the corresponding revisions made to the manuscript. All changes in the revised version are highlighted in red for your convenience.

Comments 1:   Page 5, top, “A prediction motif analysis was performed…” It sounds confusing for what is being predicted. 

Response 1:

Thank you for pointing out this problem. We agree with that “A prediction motif analysis was performed…”  has the defect of semantic ambiguity, which indeed easily confuses readers about the analyzed object and the predicted target. The original sentence only mentioned "predictive motif analysis", without explicitly stating that the analysis object was the amino acid sequence of the protein, nor did it elaborate on the specific content and significance of the prediction. We are deeply aware that in scientific papers, the description of key methods must be clear and complete to ensure that readers can accurately understand the research logic and technical path.

We have made the following targeted modifications to the original text: In the "Results" section2.2, the original sentence “A prediction motif analysis was performed…” was modified as follows: “A protein motif prediction analysis was performed using MEME-Suite in order to identify the conserved sequence patterns present within the LvCHH sequence (Figure 2A). To predict the function of LvCHH, its protein motifs were compared to the known functional protein motifs contained within the Expasy Prosite database (Figure 2B) used the Motif Comparison Tool (tomtom) in MEME-suite. These conserved motifs play pivotal roles in determining protein structure, function, and interaction properties. By precisely predicting these motifs, we can deduce the potential functions of target proteins, understand their participation in biological pathways, and lay a theoretical foundation for subsequent experimental validations.”

Comments 2:   Page 5, Figure 2, more details are expected for the legend. For example, what does the upper and lower panel represent in 2B? What does the error bar represent? And the legend of 2B (“CHH motif functional sites”) is too abstract and doesn’t reflect the figure (sequence logo representation).

Response 2:

We fully agree with your opinion. There are problems such as missing key information and ambiguous expression in the legend of Figure 2. The original legend does not clearly define the content of the upper and lower figures and the meaning of the error bars, which is very likely to cause difficulties for readers to understand. For this, we thank you for pointing out the problem in time. And we made the following modifications:

  1. Modify the original Figure 2B legend "CHH motif functional sites" to "Sequence logo and alignment analysis of CHH protein motif functional sites", which more accurately reflects the sequence identification map and comparison analysis of the CHH motif functional sites in the figure content. In addition, it is supplemented and explained in the legend of Figure 2B: "The lower figures take the predicted CHH protein motifs as the query motif. The upper figure shows the target motifs that were compared. The figure 2B shows the optimal alignment of the two motifs.”
  2. The meaning of the error bar:Toggle error bars indicating the confidence of a motif based on the number of sites used in its creation.

Comments 3: Page 6, top, “To further investigate the classification and functional characterization…” The phylogenetic analysis is just inference. It doesn’t provide any “functional” evidence. The language “functional characterization” is overloaded. 

Response 3:

Thank you for your suggestion. Phylogenetic analysis mainly constructs evolutionary relationships based on sequence similarity. Essentially, it is an inference based on sequence information and indeed cannot directly provide functional evidence. We have made the following revisions: In the "Results" section2.2 (page 6), the original sentence “To further investigate the classification and functional characterization of the CHH gene family...” was modified as follows: “To further investigate the classification and evolutionary relationships of the CHH gene family... ” Clearly stating that the purpose of this analysis is to explore the classification and evolutionary relationships and avoid over-interpretation.

Comments 4:    Page 6, “To verify whether CHH Ia is a key gene…” Here, the authors only showed the TFBS of CHH Ia genes are likely to be enriched in glucose metabolism pathways, but no quantitative analysis. As this study does not gather evidence that these TFBS are indeed open chromatin and bound by glucose metabolism transcription factors, it is not sufficient to claim that CHH Ia genes are under regulatory control of glucose metabolism. Therefore, the authors recommended that they make a comparison to a background scenario where random genes are selected. 

Response 4:

It has to be admitted that this prediction lacks experimental evidence. Thank you very much for your valuable suggestions. In the early stage of the experiment, we intended to predict the key CHH gene for glucose metabolism through various bioinformatics analysis methods. In this article, we only take the prediction of transcription factor binding sites as a method for screening the target genes. Regarding the specific functions of the genes, we verified the functions of the genes we screened by using RNAi interference and recombinant protein injection experiments.

In order to make the evidence of the result more sufficient, heatmap was generated using the FPKM values of the aforementioned transcription factors and randomly selected background transcription factors based on transcriptomic data. Among these, only CREB exhibited a decrease in expression upon gene interference, suggesting that CREB may serve as a key transcription factor involved in CHH-mediated regulation of glucose metabolism.  The relevant content has been modified in the text.

Comments 5:    Figure 5A, how is the “relative expression” quantified? Is it from PCR or RNA-seq? The Methods (Section 4.6) suggest it was “FPKM values” but not very clear here. 

Response 5:

We need to clarify that Figure 5A and Figure 5B employ two distinct methods for analyzing gene expression patterns. Specifically, Figure 5A is based on transcriptome data from a previously published article by our laboratory ([71] Zhang et al., 2019), which utilized RNA-Seq to obtain the FPKM (Fragments Per Kilobase of transcript per Million mapped reads) values of CHH genes in different tissues of L. vannamei. The expression heatmap was generated using log2-transformed and normalized data derived from these FPKM values. FPKM is a widely used metric in RNA-Seq analysis to quantify gene expression levels, representing the number of fragments per kilobase of transcript length per million mapped reads. In Figure 5B, we utilized qPCR (quantitative polymerase chain reaction) experiments to investigate the tissue expression patterns of the gene. This approach allows for a precise quantification of gene expression levels across different tissues.

About the “relative expression”, in Figure 5A, FPKM is a method used to quantify gene expression levels in RNA-seq experiments. It takes into account two key factors—sequencing depth and transcript length—to enable comparability of gene expression levels across different samples. In Figure 5B,gene expression analysis took 18S rRNA as the internal reference and used the 2−∆∆Ct method to calculate the relative expression level of the target gene in different tissues, which was mentioned in the method

Since the original result image in Figure 5A was not data normalized, we have made corrections to this figure. The modified figure is as follows:

Comments 6: Page 10, bottom, “All raw reads were deposited at the NCBI Sequential Read Archive (SRA) (PRJNA1240581)” This dataset cannot be found by the said accession number (https://www.ncbi.nlm.nih.gov/search/all/?term=PRJNA1240581). Could the authors double check? 

Response 6:

Upon checking, the reason why you might not be able to find the dataset using the provided accession number on the regular NCBI search is that the dataset is currently in the "To be released" status with a scheduled release date of 2026 - 03 - 31. Before the release date, it is not publicly accessible through the normal search channels on NCBI. Therefore, we have applied for a temporary access link. You can view the relevant data through the following link: https://dataview.ncbi.nlm.nih.gov/object/PRJNA1240581?reviewer=44h4sovm18turheua5jr79gh97. Once the dataset is released on the specified date, it should be retrievable using the accession number PRJNA1240581. We apologize for any confusion this may have caused and will ensure to clarify such details more clearly in our manuscript to avoid similar misunderstandings in the future.

Comments7:   Figure 7C-F, the barplots have illegible x-labels. The authors are expected to improve the figure resolution.   

Response 7:

Thank you very much for pointing out the problems existing in Figure 7C-F. We fully accept your suggestion and have reprocessed the image, significantly enhancing the graphic resolution to ensure that the X-axis labels are clearly distinguishable. At present, we have replaced the image version with a higher quality one to ensure that no similar problems will occur in the subsequent review and publication process. Thank you again for your detailed review and valuable suggestions, which will be of great help to us in improving this manuscript. 

We deeply appreciate your contributions and hope that the revised manuscript meets your expectations.

With best regards, 

The Authors 

Round 2

Reviewer 1 Report

Comments and Suggestions for Authors

I believed the paper had been revised adequately and can be considered for publication.

Reviewer 2 Report

Comments and Suggestions for Authors

The authors have revised the manuscript and most of my concerns are addressed. I don't have more questions now.